# Black Box Causal Inference: Effect Estimation via Meta Prediction

## Abstract

Causal inference and the estimation of causal effects plays a central role in decision-making across many areas, including healthcare and economics. Estimating causal effects typically requires an estimator that is tailored to each problem of interest. But developing estimators can take significant effort for even a single causal inference setting. For example, algorithms for regression-based estimators, propensity score methods, and doubly robust methods were designed across several decades to handle causal estimation with observed confounders. Similarly, several estimators have been developed to exploit instrumental variables (IVs), including two-stage least-squares (TSLS), control functions, and the method-of-moments. In this work, we instead frame causal inference as a dataset-level prediction problem, offloading algorithm design to the learning process. The approach we introduce, called black box causal inference (BBCI), builds estimators in a black-box manner by learning to predict causal effects from sampled dataset-effect pairs. We demonstrate accurate estimation of average treatment effects (ATEs) and conditional average treatment effects (CATEs) with BBCI across several causal inference problems with known identification, including problems with less developed estimators.

## 1 Contribution

Causal effect estimation is a fundamental problem in decision-making across many domains, like healthcare and economics. The goal in effect estimation is to estimate the effect of an intervention (e.g., a treatment) on an outcome. In order to estimate causal effects, the first step is to make *identification assumptions*, under which the desired causal effect could be uniquely determined from infinite data (Pearl et al., 2009). But, even after identification assumptions are met, we must still choose — or we may even have to develop — an *estimator* that can then be used to infer the desired effect in practice, given only our finite observed data. In this work, we consider how computation can help simplify and improve estimator development: to lessen the amount of analytical effort that causal estimation requires, we propose *learning to estimate* causal effects.

Typically, each identification setting requires a different type of estimator, and immense effort and time can go into building and improving such estimators for even a single identification setting. For example, regression-based estimators for the observed confounding setting date back to at least the 1970s (Rubin, 1974), with Wright (1921) showing special cases, followed by inverse-propensity weighting (IPW) estimators in the 1980s (Rosenbaum & Rubin, 1983). Deriving their combinations — augmented inverse-propensity weighting (AIPW) (Scharfstein et al., 1999) and targted maximum likelihood estimation (TMLE) (Van Der Laan & Rubin, 2006) estimators — occurred some 20 years later. A similar trajectory holds for causal estimation with IVs. Estimators developed for the IV setting range from two-stage least-squares (TSLS) from the 50s (Basmann, 1957) and control functions from the 70s (Heckman, 1976) to double machine learning (Chernozhukov et al., 2018) and autoencoders (Puli & Ranganath, 2020), each occurring 30 years later. As a final example, estimation with proxy variables started with negative controls (Rosenbaum, 1989) and was later further developed in the 2010s (Miao et al., 2018; Tchetgen et al., 2020). Estimators that work with proxy variables range from matrix inversion to solving an integral equation (Miao et al., 2018), and new methods are still being developed (Liu et al., 2024). The time it took to develop the above estimators speaks to the immense analytical effort that can go into deriving algorithms for each causal inference setting.

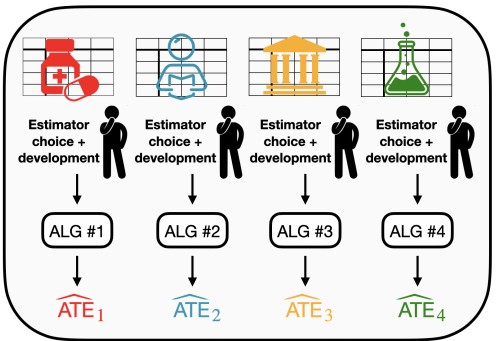 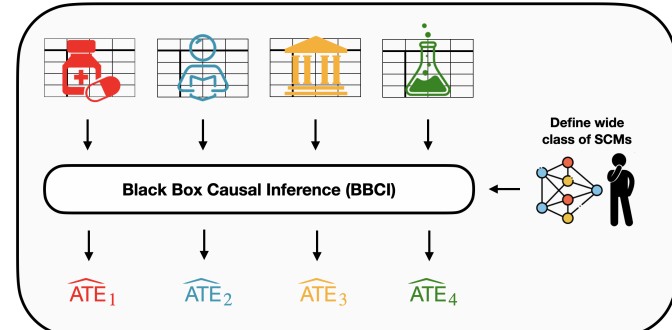

Figure 1: Visual depiction of per-dataset algorithm development (left) compared to BBCI (right).

Beyond settings with known estimators, there are also several causal inference settings that do not yet have well-established estimators. This is especially true for heterogeneous datasets where different subsets fit distinct identification assumptions. For example, there is no consensus on how to estimate effects from a dataset containing data from both a small randomized control trial and a large observational study. Moreover, even when estimators do exist, finding a *good* estimator takes considerably more effort (Robins, 2000).

We propose to use meta-learning to simplify estimator development, both in settings with known estimators and in settings without well-established estimators, leading us to the following problem statement.

**Problem Statement.** *Given a causal estimation task with covariates $\tilde{\mathbf{x}}$, treatment $\tilde{t}$, outcome $\tilde{y}$, and a causal query $\tilde{q}$ that is identifiable from dataset $\tilde{D} = \{(\tilde{\mathbf{x}}, \tilde{t}, \tilde{y})_i\}_{i \leq N}$, can we learn an effect estimation algorithm to predict $\tilde{q}$ from $\tilde{D}$?*

**Approach.** We define black box causal inference (BBCI) as a meta-prediction approach to this problem. Assume there is a true structural causal model (SCM) or class of SCMs $\mathcal{F}_0$ that defines the desired mapping $\phi_0 : \tilde{D} \mapsto \tilde{q}$. Query $\tilde{q} = \phi_0(\cdot)$ can be viewed as a function $\phi_0$ parameterized by a given SCM $S$. The class $\mathcal{F}_0$ is unknown. To restate our problem in different terms, *can we specify an SCM family $\mathcal{F}$ — some distribution over SCMs $S \sim \mathcal{F}$ — that will allow us to learn $\phi_0$?* BBCI learns a treatment effect estimator by defining an appropriate SCM family $\mathcal{F}$, simulating dataset-effect pairs $\{(\tilde{D}, \tilde{q})\}$ from samples $S \sim \mathcal{F}$, and learning to predict $\tilde{q}$ from $\tilde{D}$ in a supervised fashion.

We introduce BBCI in Section 2 as a method to learn to estimate causal effects in any setting where identification holds. We decompose the error in effect estimation into four components: uncertainty due to finite data, finite sample error, estimator variance, and error due to lack of identification (Theorem 1). We show how BBCI's error is driven to zero only when identification holds and the correct effect is estimated well across the training process. We then demonstrate in several experiments how our procedure is able to recover the target estimand $\phi_0$ across the following settings of interest.

## 1.1 Recovering $\phi_0$ when $\mathcal{F}_0$ and $\mathcal{F}$ are the same

We first find that we are able to successfully recover $\phi_0$ when $\mathcal{F}_0$ and $\mathcal{F}$ are the same. In such cases, we define $\mathcal{F}$ by clamping on a particular causal structure and a prior distribution $p(\tilde{\mathbf{x}}, \tilde{t}, \tilde{y})$. We demonstrate in Section 3 that BBCI can accurately learn to predict ATEs and CATEs across several causal inference problems with known identification, including observed confounding, IVs, and proximal causal inference, as well as problems with less developed estimators, like estimation with mixed identification conditions.

Across each of these cases, the same BBCI approach often outperforms existing baselines that were derived for each specific setting. This is useful for the development of efficient estimation algorithms for settings with known identification, which otherwise can require significant manual work or even decades of research.

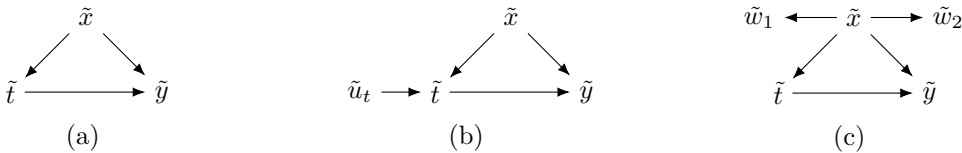

Figure 2: Example DAGs for several settings with known identification of the effect of $\tilde{t}$ on $\tilde{y}$, including: (a) the confounding case with observed confounder $\tilde{x}$; (b) the IV case with instrument $\tilde{u}_t$, where $\tilde{x}$ is unobserved; and (c) a proximal causal inference case with two proxies, $\tilde{w}_1$ and $\tilde{w}_2$.

## 1.2 Recovering $\phi_0$ when $\mathcal{F}_0$ and $\mathcal{F}$ are different

Next, we find that there are cases we can successfully recover $\phi_0$ even when $\mathcal{F}_0$ and $\mathcal{F}$ are different. Specifically, we consider cases where $\mathcal{F}_0$ and $\mathcal{F}$ are clamped on the same causal structure, the same $p(\tilde{\mathbf{x}})$, and the same $p(\tilde{t} \mid \tilde{\mathbf{x}})$, but differ in $p(\tilde{y} \mid \tilde{\mathbf{x}}, \tilde{t})$. Building on Section 3, Section 4 shows cases where BBCI successfully estimates treatment effects when $\mathcal{F}_0 \neq \mathcal{F}$ due to different types of response surfaces (e.g., spline-generated outcomes, tree-generated outcomes, and multi-layer perceptron (MLP)-generated outcomes). In another case, we show success when $\mathcal{F}_0 \subseteq \mathcal{F}$, where BBCI trained with multiple types of outcome surfaces performs well on each type individually.

## 1.3 Recovering $\phi_0$ when $\mathcal{F}_0$ is real and unknown

Finally, we demonstrate in Section 5 cases where BBCI is able to recover $\phi_0$ when $\mathcal{F}_0$ is real and unknown, using real data. We show that clamping on a particular causal structure, using the real observed dataset to define $p(\tilde{\mathbf{x}})$ and $p(\tilde{t} \mid \tilde{\mathbf{x}})$, and manually simulating $p(\tilde{y} \mid \tilde{\mathbf{x}}, \tilde{t})$ allows us to successfully recover $\phi_0$ with the LaLonde dataset in the presence of observed confounding (Lalonde, 1984; Dehejia & Wahba, 1999). BBCI outperforms per-dataset baselines, and, especially when the dataset size is smaller, produces lower variance estimates that line up with the ATE from a corresponding randomized control trial.

# 2 Black box causal inference (BBCI)

**Notation.** We use the SCM framework (Pearl et al., 2009) to formalize causal inference and define an SCM as a tuple $S = (\mathbf{x}, \mathbf{u}, \mathbf{f}, P_{\mathbf{u}})$ over a set of observed variables $\mathbf{x}$ and exogenous variables $\mathbf{u}$. In an SCM, exogenous variables follow distribution $P_{\mathbf{u}}$, and each $\tilde{x}_i$ (the $i$th component of $\mathbf{x}$) is defined by a deterministic function $f_i \in \mathbf{f}$ taking as input exogenous variable $\tilde{u}_i \in \mathbf{u}$ and the set of endogenous variables $\mathbf{x}_{\mathrm{pa}_i} \subseteq \mathbf{x}$ that directly cause $\tilde{x}_i$, i.e., $\tilde{x}_i = f_i(\mathbf{x}_{\mathrm{pa}_i}, \tilde{u}_i)$. Throughout the experiments in the paper, we consider several settings with known identification, each depicted in Figure 2. We use $\tilde{y}, \tilde{t}, \tilde{x}, \tilde{u}_t, \tilde{w}$ to denote the outcome, treatment, confounders, IVs, and proxies respectively across different data generating processes (DGPs). Across these DGPs, we consider average and conditional average treatment effects as the target $\phi_0$, each defined as averages of the outcome surface $f_y(\tilde{x}, \tilde{t}, \tilde{u}_y)$ under the interventional distributions where $\tilde{t}$ is intervened on to be 1 and 0:

$$\mathrm{CATE}(x) = \mathbb{E}_{p(\tilde{u}_y)} \left( f_y(x, \mathrm{do}(\tilde{t} = 1), \tilde{u}_y) - f_y(x, \mathrm{do}(\tilde{t} = 0), \tilde{u}_y) \right), \qquad \mathrm{PATE} = \mathbb{E}_{p(\tilde{x})} \mathrm{CATE}(\tilde{x}).$$

## 2.1 Illustration in the IV setting

We motivate BBCI with an illustration in the IV setting. IV methods aim to estimate a treatment effect in the presence of hidden confounding by isolating the variation in the treatment due to a secondary variable that effects the outcome only through the treatment, termed an instrument or instrumental variable (IV).

Consider estimating the population average treatment effect (PATE) $\phi_{\mathrm{PATE}}$ from dataset $D_0 = \{y_i, t_i, u_{t,i}\}_{i \leq N}$ with scalar continuous outcome $\tilde{y}$, treatment $\tilde{t}$, and instrument $\tilde{u}_t$. We call $D_0$ the *source dataset.* To satisfy identification, imagine a practitioner has decided to assume that the outcome function and the treatment function are both linear meaning. To estimate the PATE, the standard approach they may use is the following: (1) find out if the IV is weak; (2) test for noise heteroscedasticity; and then (3)

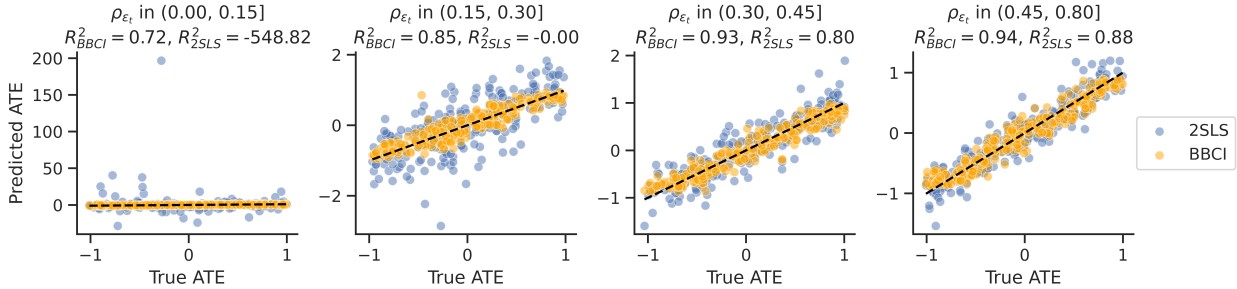

Figure 3: Predicted ATE values from the two approaches for various strengths of the instrument on datasets of size 100. Instrument strength $\rho_{\epsilon_t}$ is measured as absolute correlation between the instrument and treatment. BBCI performs as well or better than TSLS regardless of instrument strength. It is strictly better when the IV is weak, where TSLS estimates can vary wildly.

choose between different existing estimators like TSLS, the control function method, and the method of moments.

We propose an alternate approach called **black box causal inference (BBCI)** that will instead *learn* to estimate the PATE. For dataset $D_0$, BBCI can be broken down into the following procedure.

1. Define an SCM-sampler $\mathcal{F}$ that describes dataset $D_0$ under the assumption that the outcome function and treatment function are linear meaning. For example, define a given SCM $S_\nu$ as a joint distribution $p_\nu(\tilde{x}, \tilde{u}_t, \tilde{u}_y)$ and values of linear coefficients $\gamma_x, \gamma_t, \beta_x, \beta_t, \beta_y$ in the outcome and treatment functions $f_y, f_t$:

$$S_\nu \equiv \begin{cases} \tilde{x}, \tilde{u}_t, \tilde{u}_y \sim p_\nu \\ \tilde{t} = f_t(\tilde{x}, \tilde{u}_t) = \gamma_x \tilde{x} + \gamma_t \tilde{u}_t \\ \tilde{y} = f_y(\tilde{x}, \tilde{t}, \tilde{u}_y) = \beta_x \tilde{x} + \beta_t \tilde{t} + \beta_y \tilde{u}_y \end{cases}$$

   where, for example, $\tilde{x}, \tilde{u}_t, \tilde{u}_y, \gamma_x, \gamma_t, \beta_x, \beta_t, \beta_y \sim \mathcal{U}(-1, 1)$.

2. Randomly sample a parameterized SCM $S_\nu \sim \mathcal{F}$. This describes a possible DGP of interest.

3. Sample from $S_\nu$ a dataset of size $N$ of outcome-treatment-IV triples, i.e., sample $\tilde{D} = \{y_j, t_j, u_{t,j}\}_{j \leq N}$.

4. Estimate the PATE by forward sampling a large enough number $K$ times through $f_y$ and averaging over samples of confounder $\tilde{x}$ and outcome noise $\tilde{u}_y$: $\phi_{\text{PATE}}(K; S_\nu) = \frac{1}{K} \sum_{k=1}^{K} \left( f_y(x_k, \text{do}(\tilde{t} = 1), u_{y,k}) - f_y(x_k, \text{do}(\tilde{t} = 0), u_{y,k}) \right)$.

5. For a chosen model class $f_\theta$, repeat steps (3) and (4) as many times as necessary to minimize mean squared error (MSE):

$$\theta_K^* \leftarrow \arg\min_\theta \mathbb{E}_{S_\nu \sim \mathcal{F}} \mathbb{E}_{\tilde{D} \sim S_\nu} \left( \phi_{\text{PATE}}(K; S_\nu) - f_\theta\left(\tilde{D}\right) \right)^2. \tag{1}$$

6. After minimizing Equation (1), compute $f_{\theta_K^*}(D_0)$ and return it as the PATE estimate.

We compare the above procedure using a Set Transformer++ (ST++) (Zhang et al., 2022) as our model class $f_\theta$ against the well-established TSLS procedure in Figure 3. In this example, we consider source datasets $D_0$ that come from the same SCM-sampler used during training, i.e., $\mathcal{F}_0 = \mathcal{F}$. TSLS is fit on each sampled dataset individually, whereas all predictions from BBCI come from one forward pass of the same trained model. Figure 3 shows that BBCI performs as well or better than TSLS regardless of instrument strength. It is strictly better when the IV is weak, where TSLS estimates can vary wildly.

---

**Algorithm 1** Black box causal inference (BBCI) to estimate the PATE for an observed dataset $D_0$

---

**Input:** source dataset $D_0$ of size $N$, SCM-sampler $\mathcal{F}$, target estimand $\phi$, estimand sample size $K$, chosen model class $f_\theta$, batch size $m$

1: $\theta \leftarrow$ Initialize parameters
2: **for** number of training iterations **do**
3:     $\mathbf{b} \leftarrow$ Instantiate training batch of size $m$
4:     **for** $i = 1, \dots, m$ **do**
5:         Sample an SCM $S_i \sim \mathcal{F}$ from SCM-sampler $\mathcal{F}$
6:         Sample a dataset $\tilde{D}_i \sim S_i$ of size $N$ from $S_i$
7:         Use $S_i$ and estimand sample size $K$ to compute $\phi(K; S_i)$
8:         $\mathbf{b}[i] \leftarrow (\tilde{D}_i, \phi(K; S_i))$
9:     **end for**
10:     Update $f_\theta$ via gradient descent using the MSE of $\phi(K; S_i)$ predictions across batch $\mathbf{b}$:

$$\nabla_\theta \tfrac{1}{m} \sum_{i=1}^{m} \left( \phi(K; S_i) - f_\theta\left(\tilde{D}_i\right) \right)^2.$$

11: **end for**
12: **return** $f_\theta(D_0)$

---

With identifiability holding, $\lim_{K \to \infty} \phi_{\text{PATE}}(K; S_\nu) \to \beta_t$. Imagine Equation (1) is solved perfectly with MSE driven to 0. Then, other than a measure zero set of values of $\beta_t \in \{-1, 1\}$, the model recovers $\beta_t$ correctly. Formally, for any $\epsilon > 0$, with a sampling distribution over the data $\tilde{D}$ and parameterized SCMs $S_\nu$, if loss from Equation (1) converges to 0 as $K \to \infty$, by Markov's inequality, we have

$$\lim_{K \uparrow \infty} p_{\beta_t, \tilde{D}} \left( \left| \beta_t - f_{\theta_K^*}(\tilde{D}) \right|^2 > \epsilon \right) \leq \lim_{K \uparrow \infty} \frac{\mathbb{E}_{\beta_t, \tilde{D}} \left( \left| \beta_t - f_{\theta_K^*}(\tilde{D}) \right|^2 \right)}{\epsilon} \to 0.$$

Thus, if $D_0$ is a dataset sampled from an SCM in $\mathcal{F}$ with identifiability, then $f_{\theta_K^*}(D_0)$ will converge to the true causal effect.

### 2.2 Toward a "black-box" causal inference algorithm

For a practitioner who wants to estimate effects on some dataset, BBCI alleviates both the burden of finding estimators for each new setting and, if estimators already exist, choosing the best one. For instance, in the above example, if the IV is weak or the noise is heteroscedastic, using TSLS could produce biased or prohibitively high variance estimates, and another estimation algorithm would need to be chosen. BBCI is a template for a "black-box" causal inference algorithm because it:

- does not restrict the type of variables in the observed dataset to be used in estimation;
- does not require analytic derivation based on the observed data/identification assumptions; and
- does not change, or needs minimal changes, to work with different identification assumptions or observed data settings.

In other words, BBCI eschews the practice of deriving estimators for each individual problem and frames causal inference as a meta-learning problem, offloading algorithm design to the learning process. The motivation behind BBCI is to not have to derive new estimators for different identification conditions in the way it has been done so far, for conditions like separable outcome functions or invertible treatment functions (Puli & Ranganath, 2020), causal redundancy (Puli et al., 2020), or completeness (Tchetgen et al., 2020). The general BBCI recipe is described in Algorithm 1.

**Generic estimands in BBCI.** Another complication that occurs in choosing causal effect estimators is that methods developed for one estimand are not always able to estimate another. For example, estimators for ATEs often cannot estimate conditional treatment effects, where prediction involves conditioning on

a set of covariates. For example, AIPW and TMLE were constructed for ATE estimation and do not estimate CATE. BBCI, by contrast, can easily be adapted for conditional estimands like CATE. To estimate conditional estimands with BBCI, we need only sample a *query point* along with each dataset-effect pair that specifies the value to condition on in the estimand. For example, for CATE conditional on confounder setting $\tilde{x} = x$, the estimand is computed as

$$\phi_{\text{CATE}}(x, K; S) = \frac{1}{K} \sum_j \left( f_y(x, \text{do}(\tilde{t} = 1), u_{y,j}) - f_y(x, \text{do}(\tilde{t} = 0), u_{y,j}) \right).$$

Similarly, for the sample average treatment effect (SATE) given an observed dataset $D$ with $N$ units,

$$\phi_{\text{SATE}}(D; S) = \frac{1}{N} \sum_{i=1}^{N} \left( f_y(x_i, \text{do}(\tilde{t} = 1), u_{y,i}) - f_y(x_i, \text{do}(\tilde{t} = 0), u_{y,i}) \right).$$

Rather than describing the overall population from SCM $S$ as the PATE does, the sample average treatment effect (SATE) is instead specific to the $N$ observations in $D$. We emphasize the distinction in each case by specifying the relevant arguments to function $\phi$.

**Permutation-invariant prediction.** To minimize Equation (1) in the BBCI recipe, we need to specify a model class $f_\theta$. Many important causal estimands such ATEs, ATTs, CATEs, and LATEs are permutation-invariant functions of the data. Leveraging this insight, we work with permutation-invariant architectures; in other words, changing the order in which the samples appear in the data does not change the output of the model. Specifically, we use the Set Transformer++ (ST++) (Zhang et al., 2022) and increase depths and attention heads as needed to handle inference in larger datasets.

## 2.3 Causal error decomposition

To shed light on what BBCI optimizes for, we decompose BBCI's error into four constituents: Monte Carlo error, coming from sample-based estimates of the target estimand; model error, comparing $f_\theta$ to the optimal predictor given finite data; identification error, arising from the non-identifiability of the causal effect; and finite-sample error due to our use of finite dataset sizes.

**Theorem 1.** *Let $\phi(S)$ be the true target quantity of interest under a given SCM $S \sim \mathcal{F}$. $\tilde{\phi}(S)$ approximates $\phi(S)$ under a finite resource constraint, such as a finite number of samples for Monte Carlo estimation. Assume $\phi(S) = \tilde{\phi}(S) + \epsilon$, where $\mathbb{E}[\epsilon] = 0$ with $\epsilon$ independent and finite variance. Let $\tilde{D}_N$ be a dataset from which to predict $\phi(S)$ and assume a measure space exists such that $\tilde{D}_\infty = \lim_{M \to \infty} \tilde{D}_M$ is well defined. Assume a sampling distribution over $\phi(S), \tilde{\phi}(S), \tilde{D}_N, \tilde{D}_\infty$. The training error for BBCI is then*

$$\mathbb{E}\left[ \left( \phi(S) - f_\theta(\tilde{D}) \right)^2 \right] = \underbrace{\mathbb{E}\left[ \left( \phi(S) - \tilde{\phi}(S) \right)^2 \right]}_{\substack{\text{A. MC error in the} \\ \text{targeted causal parameter}}} + \underbrace{\mathbb{E}\left[ \left( f_\theta(\tilde{D}_N) - \mathbb{E}\left[ \phi(S) \mid \tilde{D}_N \right] \right)^2 \right]}_{\substack{\text{B. model error from the} \\ \text{optimal predictor given } N \text{ samples}}}$$

$$+ \underbrace{\mathbb{E}\left[ \left( \phi(S) - \mathbb{E}\left[ \phi(S) \mid \tilde{D}_\infty \right] \right)^2 \right]}_{\substack{\text{C. identification error (uncertainty} \\ \text{in the causal estimand given the best} \\ \text{predictor with an infinite dataset)}}} + \underbrace{\mathbb{E}\left[ \left( \mathbb{E}\left[ \phi(S) \mid \tilde{D}_\infty \right] - \mathbb{E}\left[ \phi(S) \mid \tilde{D}_N \right] \right)^2 \right]}_{\substack{\text{D. finite sample error} \\ \text{of the optimal predictor}}}.$$

The proof is in Appendix A. The four error terms A, B, C, and D together account for the MSE in BBCI. The first term, A, arises from the fact that we use sample-based estimates for the target quantity given an SCM. By using more resources at the time of data creation, we can reduce this quantity. Term B captures model error relative to the optimal predictor from $N$ samples. The third term, C, arises from non-identifiability due to uncertainty in the causal estimand itself. Even when we use an infinitely large dataset drawn from $S$, this term cannot be reduced further. Finally, finite-sample error in term D is due to our use of a finite-sized dataset $\tilde{D}_N$. We try to approximate the estimate we would get from an infinitely large dataset $\tilde{D}_\infty$.

# 3 Experiments when $\mathcal{F}_0 = \mathcal{F}$

In this section, we consider cases where $\mathcal{F}_0 = \mathcal{F}$. We demonstrate accurate effect estimation across several causal inference problems with known identification, including settings with confounders, instrumental variables, and proxies. We also validate our method on the Infant Health and Development Program (IHDP) dataset with real covariates and known effects (Hill, 2011). We base our simulations for known identification settings on the following DGP in Equation (2).

Across the confounding, IV, and proxy experiments with DGP (2), we target the PATE, which we define for a continuous treatment as the effect of a one-unit increase in $t$ (averaged across the range of possible $t$ values), and in the binary case as the effect of a one-unit increase in $t$ from 0 to 1 (using binary $t \sim \text{Bern}(\sigma(\gamma_x \tilde{x} + \gamma_t \tilde{u}_t))$ with sigmoid $\sigma$).

$$
\begin{aligned}
\gamma_t, \beta_t, \beta_y &\sim \mathcal{U}(-1, 1) & \tilde{u}_y^{(i)}, \tilde{u}_t^{(i)} &\sim \mathcal{N}(0, 1) & \tilde{w}_2^{(i)} &\sim \mathcal{N}(\tilde{x}^{(i)}, \delta_2) \\
\gamma_x, \beta_x &\sim \mathcal{U}(-2, -1) \cup \mathcal{U}(1, 2) & \tilde{x}^{(i)} &\sim \mathcal{N}(0, 1) & \tilde{t}^{(i)} &= \gamma_x \tilde{x}^{(i)} + \gamma_t \tilde{u}_t^{(i)} \\
\delta_1, \delta_2 &\sim \mathcal{U}(0, 1) & \tilde{w}_1^{(i)} &\sim \mathcal{N}(\tilde{x}^{(i)}, \delta_1) & \tilde{y}^{(i)} &= \beta_x \tilde{x}^{(i)} + \beta_t \tilde{t}^{(i)} + \beta_y \tilde{u}_y^{(i)}
\end{aligned}
\tag{2}
$$

**Baselines for comparison.** Each causal inference setting has its own set of identification assumptions and, often, a corresponding variety of estimator choices given those identification assumptions. Because the same BBCI approach can support multiple types of settings across different sets of identification assumptions, there is no obvious choice of baseline for comparison against BBCI across settings. As a result, we compare the performance of BBCI in each case to that of both general and setting-specific baselines. We consider three baselines in Table 1, each doing per-dataset regression that is fit on each test point. The baselines successively encode increasing levels of knowledge about DGP 2. T-Only-MLP fits an MLP on each dataset using only the treatment for a naive estimate of the ATE. Reg-MLP, 2SLS-MLP, and Pr2SLS-MLP use an MLP with additional knowledge of the corresponding setting in a manner that satisfies identification (e.g., including the confounder or performing two-stage least-squares in the appropriate manner). Reg-Lin, 2SLS-Lin, and Pr2SLS-Lin do the same, but assuming additional knowledge of the linear functional forms in DGP (2). We briefly discuss each setting individually.

Table 1: PATE estimation for continuous treatment settings with DGP (2). Possible PATE values range from -1 to 1.

| Setting | Model | $R^2$ | | RMSE | |
| | | N=100 | N=1000 | N=100 | N=1000 |
| --- | --- | --- | --- | --- | --- |
| Naive | T-Only-MLP | 0.2526 | 0.2729 | 0.9778 | 0.9453 |
| Confounder | BBCI | **0.8853** | **0.9570** | **0.1833** | **0.1172** |
| Confounder | Reg-MLP | 0.4576 | 0.7544 | 0.6250 | 0.2887 |
| Confounder | Reg-Lin | 0.1224 | 0.7070 | 1.5910 | 0.3724 |
| Instrument | BBCI | **0.8514** | **0.9394** | **0.2050** | **0.1359** |
| Instrument | 2SLS-MLP | 0.3812 | 0.6290 | 0.7419 | 0.4618 |
| Instrument | 2SLS-Lin | $\leq 0$ | 0.2309 | 43.7167 | 1.0341 |
| Proxy | BBCI | **0.8885** | **0.9473** | **0.1837** | **0.1224** |
| Proxy | Pr2SLS-MLP | 0.3728 | 0.3711 | 0.7035 | 0.7135 |
| Proxy | Pr2SLS-Lin | 0.1224 | 0.7070 | 1.5908 | 0.3724 |

## 3.1 Confounding

In the confounding case, the model is shown treatments and outcomes along with the confounding variable: $D = \{x^{(i)}, t^{(i)}, y^{(i)}\}_{i \leq N}$. We compare BBCI using Set Transformer++ (ST++) to per-dataset regression with a multilayer perceptron (Reg-MLP) or with linear regression (Reg-Lin), appropriate for the linear

response surface in DGP (2). Models are evaluated across 1,000 test datasets, in both a smaller dataset regime (100 observations per dataset) as well as a larger dataset regime (1,000 observations per dataset). The ST++ is trained with 8 encoding layers for 200 epochs. The "Confounder" setting rows in Table 1 summarize prediction results for each method in this case, with BBCI showing stronger performance, especially in the small data regime.

## 3.2 Instrumental variables

In the IV case, the model is shown treatments and outcomes along with an instrument for the treatment, but the confounder remains hidden: $D = \{u_t^{(i)}, t^{(i)}, y^{(i)}\}_{i \leq N}$. In this case we compare BBCI using ST++ to per-dataset two-stage least-squares (TSLS), again with either an MLP (2SLS-MLP) or linear functional forms (2SLS-Lin). See Appendix B.1 for details on identification conditions for each of the TSLS baselines. In this setting, shown in the "Instrument" rows in Table 1, BBCI significantly outperforms the TSLS baselines. The linear regression baseline struggles particularly with weak instruments, as we saw in our previous example in Figure 3.

## 3.3 Proximal causal inference

In the proximal causal inference case, the model is shown two proxies for the hidden confounder alongside treatments and outcomes: $D = \{w_1^{(i)}, w_2^{(i)}, t^{(i)}, y^{(i)}\}_{i \leq N}$. We compare BBCI using ST++ to a recent regression-based approach to proximal causal inference, introduced in (Liu et al., 2024). We implement a linear regression variant (Pr2SLS-Lin) and an MLP variant of this procedure (Pr2SLS-MLP), detailed in Appendix B.2. Results in Table 1 again show BBCI working effectively in the proxy case.

## 3.4 Comparisons to other established setting-specific methods

Settings like the confounding setting and IV setting each have mature literatures focused on estimation. To see how BBCI compares to a state-of-the-art baseline in each case, we run DoubleML (DML) on both the confounder and IV settings with DGP (2). We use the partially linear regression model (PLR) for the confounder setting and the partially linear IV regression model (PLIV) for the IV setting. In each setting, we use either a linear regressor or a random forest to model the nuisance function estimators. Table 2 shows BBCI outperforming DML in the confounding setting with dataset size $N = 100$ and in the instrument setting with both dataset sizes.

Table 2: PATE estimation for continuous treatment settings, comparing BBCI to DML. Values shown after double bars ignore all predictions outside the range [-2, 2].

| Setting | Model | $R^2$ | | RMSE | |
| --- | --- | --- | --- | --- | --- |
| | | N=100 | N=1000 | N=100 | N=1000 |
| Confounder | BBCI | **0.7577 \|\| 0.7560** | 0.8088 \|\| 0.8129 | **0.2821 \|\| 0.2836** | 0.2526 \|\| 0.2498 |
| Confounder | PLR-LR | $\leq 0$ \|\| 0.7747 | 0.0230 \|\| 0.9338 | 263.25 \|\| 0.3001 | 3.6420 \|\| 0.1493 |
| Confounder | PLR-RF | 0.6761 \|\| 0.6891 | **0.8828 \|\| 0.8852** | 0.4050 \|\| 0.3846 | **0.1985 \|\| 0.1953** |
| Instrument | BBCI | **0.8597 \|\| 0.8677** | **0.9215 \|\| 0.9255** | **0.2821 \|\| 0.2836** | **0.1618 \|\| 0.1577** |
| Instrument | PLIV-LR | $\leq 0$ \|\| 0.5558 | 0.0381 \|\| 0.8118 | 6.5001 \|\| 0.4913 | 2.3702 \|\| 0.2639 |
| Instrument | PLIV-RF | $\leq 0$ \|\| 0.5240 | 0.0171 \|\| 0.8165 | 11.018 \|\| 1.5093 | 2.9379 \|\| 0.2608 |

## 3.5 Conditional average treatment effects

To test estimation in the conditional case, we consider two variations of the response surface in DGP (2) that induce treatment effect heterogeneity: $y^{(i)} = \beta_x \tilde{x}^{(i)} t^{(i)} + \beta_t \tilde{t}^{(i)} + \beta_y \tilde{u}_y^{(i)}$ and $y^{(i)} = \beta_x |\tilde{x}^{(i)}| \tilde{t}^{(i)} + \beta_t \tilde{t}^{(i)} + \beta_y \tilde{u}_y^{(i)}$. The first response surface produces CATEs that are a linear function of $\tilde{x}$, while the second produces nonlinear CATEs. Table 3 compares BBCI with ST++ to per-dataset MLP predictions $\hat{f}(t+1, x) - \hat{f}(t, x)$

Table 3: CATE estimation with nonlinear outcomes. Effects are conditional on query point $x$.

| Response (without noise) | Model | Dataset Size | $R^2$ | RMSE |
|---|---|---|---|---|
| $\beta_x \tilde{x}^{(i)} \tilde{t}^{(i)} + \beta_t \tilde{t}^{(i)}$ | BBCI | 100 | **0.9993** | **0.0407** |
| $\beta_x \tilde{x}^{(i)} \tilde{t}^{(i)} + \beta_t \tilde{t}^{(i)}$ | MLP | 100 | $\leq 0.0$ | 1.0950 |
| $\beta_x \left\| \tilde{x}^{(i)} \right\| \tilde{t}^{(i)} + \beta_t \tilde{t}^{(i)}$ | BBCI | 100 | **0.9994** | **0.0362** |
| $\beta_x \left\| \tilde{x}^{(i)} \right\| \tilde{t}^{(i)} + \beta_t \tilde{t}^{(i)}$ | MLP | 100 | 0.7629 | 9058 |
| $\beta_x \tilde{x}^{(i)} \tilde{t}^{(i)} + \beta_t \tilde{t}^{(i)}$ | BBCI | 1000 | **0.9996** | **0.0314** |
| $\beta_x \tilde{x}^{(i)} \tilde{t}^{(i)} + \beta_t \tilde{t}^{(i)}$ | MLP | 1000 | $\leq 0.0$ | 0.9220 |
| $\beta_x \left\| \tilde{x}^{(i)} \right\| \tilde{t}^{(i)} + \beta_t \tilde{t}^{(i)}$ | BBCI | 1000 | **0.9995** | **0.0357** |
| $\beta_x \left\| \tilde{x}^{(i)} \right\| \tilde{t}^{(i)} + \beta_t \tilde{t}^{(i)}$ | MLP | 1000 | 0.7835 | 0.8842 |

for each query point $x$. Note in this case, the query point $x$ is easily accommodated into the ST++ architecture simply as an additional feature. Results in the CATE case demonstrate the flexibility of BBCI in targeting conditional estimands.

### 3.6 Causal estimation on the Infant Health and Development Program (IHDP) datasets

Hill (2011) constructed the IHDP datasets using covariates and treatment from a randomized study of the impact on educational and follow-up interventions on child cognitive development. Using real covariates and treatment, they introduce confounding by removing a specific subset of the treated population, and simulating a variety of response surfaces. Instead of simulating $\tilde{x}$ and $\tilde{t}$ from DGP (2), this experiment defines $p(\tilde{x})$ and $p(\tilde{t} \mid \tilde{x})$ by *conditioning on the source dataset*. Note also in this case that $N$ for the observed data is 747 and $\tilde{x}$ is 25-dimensional. We test on both a linear and nonlinear outcome surface from (Hill, 2011) (labeled surfaces "A" and "B," respectively) and sample 13,000 training datasets and 1,000 test datasets to run BBCI. As above, we compare to a per-dataset MLP baseline. Table 4 shows BBCI estimates the SATE well when trained and tested on the same response surface across each of the settings.

Table 4: SATE estimation on IHDP data using Hill (2011) response surfaces A (linear) and B (exponential).

| Response surface | Model | $R^2$ | RMSE |
|---|---|---|---|
| Surface A (linear) | BBCI | **0.9318** | **0.1858** |
| Surface A (linear) | MLP | 0.8261 | 0.2719 |
| Surface B (exponential) | BBCI | **0.9809** | **0.1808** |
| Surface B (exponential) | MLP | 0.9070 | 0.3522 |

### 3.7 Causal estimation on mixed datasets

One of the motivating use-cases for BBCI, discussed also in Section 2, is that we can discover estimation algorithms for settings that do not yet have established estimators. To explore this further, we consider an additional DGP that mixes multiple types of data. In the "Confounder+IV" case, shown in Table 5, both a confounder and an IV are present, suggesting some combination of confounding-specific estimation methods and IV-specific estimation methods would produce a better estimator than either type of method alone. Table 5 demonstrates that BBCI is indeed able to outperform both the confounding-specific and the IV-specific baselines.

Table 5: PATE estimation for continuous treatment settings with mixed DGPs.

| Setting | Model | $R^2$ | | RMSE | |
| --- | --- | --- | --- | --- | --- |
| | | N=100 | N=1000 | N=100 | N=1000 |
| Confounder + IV (linear) | BBCI | **0.9287** | **0.9634** | **0.1524** | **0.1083** |
| Confounder + IV (linear) | T-Only-Transformer | 0.8175 | 0.9169 | 0.2438 | 0.1631 |
| Confounder + IV (linear) | Reg-MLP on Confounder | $\leq 0$ | 0.1202 | 0.8075 | 0.5307 |
| Confounder + IV (linear) | 2SLS-MLP on IV | $\leq 0$ | 0.4347 | 0.5882 | 0.4254 |

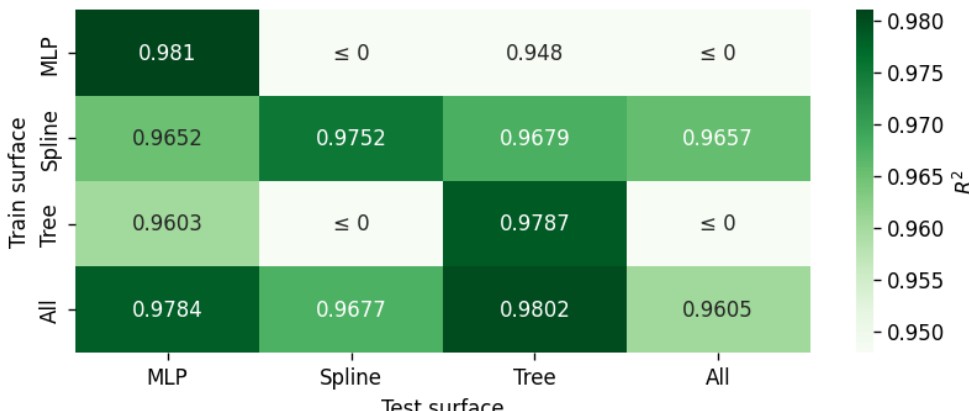

Figure 4: PATE estimates for various DGPs with binary treatment and nonlinear outcomes in the confounding case, where the target PATE is normalized by the observed outcome's 95% and 5% quantiles. The combined SCM-sampler samples from all 3 response surfaces with equal probability.

## 4 Experiments when $\mathcal{F}_0 \neq \mathcal{F}$

In this section, we extend our results in Section 3 to consider when $\mathcal{F}_0$ and $\mathcal{F}$ are different. Specifically, we consider cases where $\mathcal{F}_0$ and $\mathcal{F}$ are clamped on the same causal structure, the same $p(\tilde{x})$, and the same $p(\tilde{t} \mid \tilde{x})$, but differ in $p(\tilde{y} \mid \tilde{x}, \tilde{t})$.

We first expand the training DGP to nonlinear response surfaces $\tilde{y}^{(i)} = f(\tilde{t}^{(i)}, \tilde{x}^{(i)}) + \beta_y \tilde{u}_y^{(i)}$. Following Ke et al. (2023) and Bengio et al. (2019), we model response surfaces $f(\cdot, \cdot)$ as random trees, random MLPs, and random splines. All random MLPs have 2 layers with 10 hidden units, Leaky ReLU activation, and randomly initialized weights and biases (Ke et al., 2023). For each value of the treatment, a random spline is generated by fitting a second-order spline with $K = 8$ knots to $K$ pairs of uniformly sampled points $\{a_k^t, b_k^t\}_{k \leq K}$ in intervals $[-8, 8]$ (Bengio et al., 2019). Similarly, random trees are generated by fitting a decision tree with a maximum depth of 5. To avoid leaking information about the typical ranges of ATEs from each type of response surface, we normalize the ATE value by the observed outcome's 95% and 5% quantiles.

We compare BBCI using ST++ trained across different response surfaces and combinations of response surfaces to per-dataset Reg-MLP. Both BBCI and Reg-MLP are evaluated on the unnormalized ATEs with $N = 1000$. As shown in Figure 4, along the diagonal, BBCI predicts the PATE well when being trained and tested on the same surface. Looking at the last row, where BBCI is trained on the combined SCM-sampler that samples from all 3 response surfaces, we see that training BBCI on a larger family of response surfaces maintains its good performance on each surface individually (i.e., $\mathcal{F}_0 \subseteq \mathcal{F}$). Lastly the "Spline" row shows the results of training BBCI on splines and testing on MLP and Tree surfaces. The consistently high $R^2$ values show that it is possible for BBCI to generalize outside of the specified SCM family.

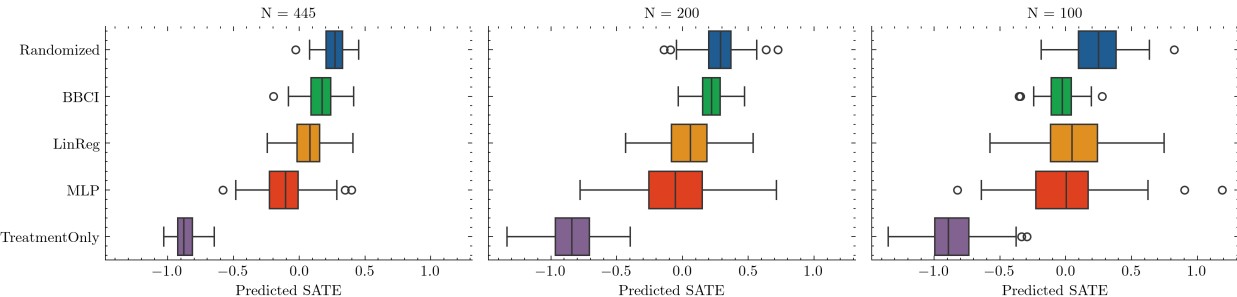

Figure 5: SATE estimates for LaLonde data (Lalonde, 1984), comparing randomized estimates of the SATE to observational estimates of the SATE across different source dataset sizes: the original study size $N = 445$ as well as two smaller subsets, $N = 200$ and $N = 100$.

## 5 Experiments when $\mathcal{F}_0$ is real and unknown

We demonstrate in this section that BBCI is able to recover target estimand $\phi_0$ when $\mathcal{F}_0$ is real and unknown, using real data. In the previous experiments, we have conditioned on particular causal structures (confounding, instruments, proxies) as well as parametric distributions, functional forms, or observed data to define covariate distributions and structural equations. In this section, we assume the confounding setting and again define $p(\tilde{x})$ and $p(\tilde{t} \mid \tilde{x})$ by conditioning on the source data.

The LaLonde dataset (Lalonde, 1984) contains 9 covariates $\tilde{x}$ for the purpose of exploring the impact of job training programs $\tilde{t}$ on later earnings $\tilde{y}$. The associated datasets from the LaLonde study as well as many followup studies (Lalonde, 1984; Dehejia & Wahba, 1999) provide both randomized experiment data, from which an estimate of the true ATE of $\tilde{t}$ on $\tilde{y}$ can be made, as well as data over the same features from additional non-participants who were not involved in the randomized experiment (i.e., observational controls), allowing for the creation of corresponding non-randomized datasets on which to test observational estimation methods.

Figure 5 shows the result of training BBCI on non-randomized LaLonde data with MLP-simulated outcomes $\tilde{y}$, and then comparing each predicted SATE to the true SATE that a randomized control trial of the same size predicts. Crucially, BBCI is trained only on MLP-simulated outcomes but is instead shown *real outcomes at test time*. This is done across three dataset sizes: the original study size $N = 445$ as well as two smaller subsets, $N = 200$ and $N = 100$. To construct each dataset during training, we first sample $N$ units from the randomized data, then replace all control units with observational control units from the much larger set of study non-participants. This creates a corresponding dataset of size $N$ that instead displays observed confounding. We then replace the outcomes $\tilde{y}$ with MLP-generated outcomes and compute a corresponding SATE according to the MLP response surface. In this manner, BBCI is shown only non-randomized data with MLP-generated outcomes during training. At test time, the same sampling procedure is used to generate observed datasets and randomized datasets of size $N$, but this time, real outcomes are used rather than simulated outcomes.

The estimates for the Randomized method in Figure 5 show SATE estimates coming from the mean difference in the randomized data, while the estimates for all other methods come instead from the observational data. LinReg and MLP use a per-dataset linear regression or per-dataset MLP to estimate the SATE, while TreatmentOnly takes a naive estimate of the SATE as the mean difference in the observational data. BBCI recovers the SATE most successfully across the three settings, with the particular benefit of lower variance estimates for smaller datasets. This result suggests BBCI can be a promising approach in real data settings.

# 6  Related work

Non-parametric identification of causal effects, achieved using graph-based criteria or functional conditions like with IVs or proxy variables, reduces causal inference to statistical inference under missing data (Imbens & Wooldridge, 2007; Pearl et al., 2009; Wang & Blei, 2019; Tchetgen et al., 2020; Puli & Ranganath, 2019; Puli et al., 2020; Fisher, 1992; Robins, 2000; Miao et al., 2018). Many works aim to establish analytical procedures for each identification setting on a case-by-case basis (e.g., Shalit et al. (2017); Kallus (2016); Wooldridge (2015); Puli et al. (2020); Van Der Laan & Rubin (2006); Miao et al. (2018)) as opposed to framing causal estimation across settings as a computational problem.

A close work to BBCI, in spirit, is (Xia et al., 2021), which proposes to search over SCMs parameterized by neural models to identify and estimate effects. The idea is to find two different SCMs that have two different interventional distributions while keeping the observed data likely (which they term consistency). For large enough data such that consistency can be checked reliably, their method relies on the expressivity of neural models to guarantee that if the two interventional distributions are the same, the effect is both identified and estimated. Their approach fails to satisfy one of the black-box criteria described in Section 2: either (1) the method needs to change based on the identification assumptions, because there is no way to encode functional assumptions like monotonicity or completeness, which are necessary for estimation with IVs and proxies; or (2) there is analytic effort involved in showing that such assumptions can be implemented into the method as-is while maintaining the expressivity that (Xia et al., 2021) require. Finally, in cases without functional assumptions, their method can be thought of as a choice of sampling strategy to refine the SCM-sampler $\mathcal{F}$ in BBCI to better represent the observed data. This maintains the black-box nature of BBCI, because the sampling strategy is a matter of implementation of $\mathcal{F}$ and does not change how the method is run, nor require derivations.

Some flexible causal discovery methods like (Geffner et al., 2022) also estimate effects as a by-product of causal discovery. This can require all observed variables in the data or can be limited to additive noise SCMs, and thus may not support the functional assumptions required for identification. We focus instead on causal estimation in this work, but the idea of predicting graph structure fits naturally into the idea of BBCI: instead of predicting causal effects from a distribution, generate the graph that fits the dependency structure of the distribution. We leave this to future work.

Müller et al. (2021) also use transformers for meta prediction, in their case doing general Bayesian inference: given a sampling distribution over prior datasets, they use a transformer to model the posterior predictive distribution by maximizing the likelihood of a test label given a test data point and a training dataset. Their work is also similar in spirit but is not a black-box algorithm for causal inference, which is a fundamentally different problem due to its focus on predicting unobserved quantities.

# 7  Discussion and future work

We frame causal inference as a dataset-level meta prediction task and propose black box causal inference (BBCI) as an approach for learning effect estimators from sampled dataset-effect pairs. We show that BBCI performs as well or better than common estimation procedures on a wide variety of DGPs and can handle estimation even in settings with less developed estimators. Finally, we validate BBCI on semi-synthetic as well as real datasets. Successful estimation in cases where $\mathcal{F}_0 \subseteq \mathcal{F}$ and $\mathcal{F}_0 \neq \mathcal{F}$ suggests a promising direction could be to specify a sampler over a wide variety and/or expressive class of SCMs. Such efforts can build on, for example, other works that specify priors over structural causal models for other related purposes Wu et al. (2024); Hollmann et al. (2022). Extending estimation to settings with higher-dimensional SCMs is another area of interest for future work. In higher dimensional spaces, specifying an SCM family $\mathcal{F}$ that induces distributions with sufficiently meaningful structure would be an additional challenge, suggesting the utility of generative methods for specifying SCM families, such as Bynum & Cho (2024); Im et al. (2024). Finally, a remaining and important question for any black box causal inference approach is to produce estimates with uncertainty. Connecting to variational and/or Bayesian methods for uncertainty quantification (Ranganath et al., 2014; Kingma & Welling, 2013; Müller et al., 2021) could extend BBCI to provide interval or distributional estimates rather than point estimates of treatment effects.

## 8 Statement of broader impact

This paper presents work whose goal is to advance the fields of Machine Learning and Causal Inference. The potential societal consequences of this work are essentially those shared by any use of meta-learning in an applied setting. A general purpose causal estimation tool like BBCI can be used in many settings, but that does not obviate the need for careful consideration of each context of use — careless and/or unthorough treatment effect estimation can ultimately lead to bad treatment advice and poor decision making.

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

# A  Proofs

**Theorem 1.** *Let $\phi(S)$ be the true target quantity of interest under a given SCM $S \sim \mathcal{F}$. $\tilde{\phi}(S)$ approximates $\phi(S)$ under a finite resource constraint, such as a finite number of samples for Monte Carlo estimation. Assume $\phi(S) = \tilde{\phi}(S) + \epsilon$, where $\mathbb{E}[\epsilon] = 0$ with $\epsilon$ independent and finite variance. Let $\tilde{D}_N$ be a dataset from which to predict $\phi(S)$ and assume a measure space exists such that $\tilde{D}_\infty = \lim_{M \to \infty} \tilde{D}_M$ is well defined. Assume a sampling distribution over $\phi(S), \tilde{\phi}(S), \tilde{D}_N, \tilde{D}_\infty$. The training error for BBCI is then*

$$\mathbb{E}\left[\left(\phi(S) - f_\theta(\tilde{D})\right)^2\right] = \underbrace{\mathbb{E}\left[\left(\phi(S) - \tilde{\phi}(S)\right)^2\right]}_{\substack{A.\ MC\ error\ in\ the \\ targeted\ causal\ parameter}} + \underbrace{\mathbb{E}\left[\left(f_\theta(\tilde{D}_N) - \mathbb{E}\left[\phi(S) \mid \tilde{D}_N\right]\right)^2\right]}_{\substack{B.\ model\ error\ from\ the \\ optimal\ predictor\ given\ N\ samples}}$$

$$+ \underbrace{\mathbb{E}\left[\left(\phi(S) - \mathbb{E}\left[\phi(S) \mid \tilde{D}_\infty\right]\right)^2\right]}_{\substack{C.\ identification\ error\ (uncertainty \\ in\ the\ causal\ estimand\ given\ the\ best \\ predictor\ with\ an\ infinite\ dataset)}} + \underbrace{\mathbb{E}\left[\left(\mathbb{E}\left[\phi(S) \mid \tilde{D}_\infty\right] - \mathbb{E}\left[\phi(S) \mid \tilde{D}_N\right]\right)^2\right]}_{\substack{D.\ finite\ sample\ error \\ of\ the\ optimal\ predictor}}.$$

*Proof.* As a first step, we extract a variance term and rewrite as the mean squared error to the true parameter.

$$\mathbb{E}\left[\left(f_\theta(\tilde{D}_N) - \tilde{\phi}(S)\right)^2\right] = \mathbb{E}\left[\left(f_\theta(\tilde{D}_N) - \phi(S) + \phi(S) - \tilde{\phi}(S)\right)^2\right]$$

$$= \mathbb{E}\left[\left(f_\theta(\tilde{D}_N) - \phi(S)\right)^2 + \left(\phi(S) - \tilde{\phi}(S)\right)^2\right] - 2\mathbb{E}\left[\left(f_\theta(\tilde{D}_N) - \phi(S)\right)\left(\phi(S) - \tilde{\phi}(S)\right)\right]$$

$$= \mathbb{E}\left[\left(f_\theta(\tilde{D}_N) - \phi(S)\right)^2 + \left(\phi(S) - \tilde{\phi}(S)\right)^2\right] - 2\mathbb{E}\left[\left(f_\theta(\tilde{D}_N) - \phi(S)\right)\epsilon\right]$$

$$= \mathbb{E}\left[\left(f_\theta(\tilde{D}_N) - \phi(S)\right)^2 + \left(\phi(S) - \tilde{\phi}(S)\right)^2\right] - 2\mathbb{E}\left[\left(f_\theta(\tilde{D}_N) - \phi(S)\right)\right]\mathbb{E}[\epsilon]$$

$$= \mathbb{E}\left[\left(f_\theta(\tilde{D}_N) - \phi(S)\right)^2 + \left(\phi(S) - \tilde{\phi}(S)\right)^2\right] - 2\mathbb{E}\left[\left(f_\theta(\tilde{D}_N) - \phi(S)\right)\right]0$$

$$= \mathbb{E}\left[\left(f_\theta(\tilde{D}_N) - \phi(S)\right)^2 + \left(\phi(S) - \tilde{\phi}(S)\right)^2\right]$$

The second term is error from targeting a noisy version of the true estimand. The next step will be to expand the first term using the traditional mean squared error. The steps are shown for clarity.

$$\mathbb{E}\left[\left(f_\theta(\tilde{D}_N) - \phi(S)\right)^2\right] = \mathbb{E}\left[f_\theta(\tilde{D}_N)^2 - 2f_\theta(\tilde{D}_N)\phi(S) + \phi(S)^2\right]$$

$$= \mathbb{E}\left[\mathbb{E}\left[f_\theta(\tilde{D}_N)^2 - 2f_\theta(\tilde{D}_N)\phi(S) + \phi(S)^2 \mid \tilde{D}_N\right]\right]$$

$$= \mathbb{E}\left[\mathbb{E}\left[f_\theta(\tilde{D}_N)^2 \mid \tilde{D}_N\right] - 2\mathbb{E}\left[f_\theta(\tilde{D}_N)\phi(S) \mid \tilde{D}_N\right] + \mathbb{E}\left[\phi(S)^2 \mid \tilde{D}_N\right]\right]$$

$$= \mathbb{E}\left[f_\theta(\tilde{D}_N)^2 - 2f_\theta(\tilde{D}_N)\mathbb{E}\left[\phi(S) \mid \tilde{D}_N\right] + \mathbb{E}\left[\phi(S) \mid \tilde{D}_N\right]^2 + \mathbb{V}ar\left[\phi(S) \mid \tilde{D}_N\right]\right]$$

$$= \mathbb{E}\left[\left(f_\theta(\tilde{D}_N) - \mathbb{E}\left[\phi(S) \mid \tilde{D}_N\right]\right)^2 + \mathbb{V}ar\left[\phi(S) \mid \tilde{D}_N\right]\right].$$

The first term is model estimation error. Note that $\theta$ could be a random variable as it is random from the training set, so we could expand it further to pull out the bias and variance from training. The latter term is however more interesting as it hides causal 'identification error,' which occurs when $\phi(S)$ is still random given an infinite dataset (assuming the existence of some causal estimator whose variance goes to zero given identification). Further expanding the last term,

$$\mathbb{E}\big[\mathbb{V}ar\big[\phi(S) \mid \tilde{D}_N\big]\big] = \mathbb{E}_{\tilde{D}_N}\left[\mathbb{E}_{\phi(S)|\tilde{D}_N}\left[\big(\phi(S) - \mathbb{E}\big[\phi(S) \mid \tilde{D}_N\big]\big)^2\right]\right]$$

$$=\mathbb{E}_{\tilde{D}_N,\phi(S)}\left[\big(\phi(S) - \mathbb{E}\big[\phi(S) \mid \tilde{D}_N\big]\big)^2\right]$$

$$=\mathbb{E}_{\tilde{D}_N,\phi(S),\tilde{D}_M}\left[\big(\phi(S) - \mathbb{E}\big[\phi(S) \mid \tilde{D}_N\big]\big)^2\right]$$

$$=\mathbb{E}_{\tilde{D}_N,\phi(S),\tilde{D}_M}\left[\big(\phi(S) - \mathbb{E}\big[\phi(S) \mid \tilde{D}_M\big] + \mathbb{E}\big[\phi(S) \mid \tilde{D}_M\big] - \mathbb{E}\big[\phi(S) \mid \tilde{D}_N\big]\big)^2\right]$$

$$=\mathbb{E}_{\tilde{D}_N,\phi(S),\tilde{D}_M}\left[\big(\phi(S) - \mathbb{E}\big[\phi(S) \mid \tilde{D}_M\big]\big)^2 + \big(\mathbb{E}\big[\phi(S) \mid \tilde{D}_M\big] - \mathbb{E}\big[\phi(S) \mid \tilde{D}_N\big]\big)^2\right.$$
$$\left. + 2\left(\phi(S) - \mathbb{E}\big[\phi(S) \mid \tilde{D}_M\big]\right)\left(\mathbb{E}\big[\phi(S) \mid \tilde{D}_M\big] - \mathbb{E}\big[\phi(S) \mid \tilde{D}_N\big]\right)\right]$$

$$=\mathbb{E}_{\tilde{D}_N,\phi(S),\tilde{D}_M}\Big[\big(\phi(S) - \mathbb{E}\big[\phi(S) \mid \tilde{D}_M\big]\big)^2 + \big(\mathbb{E}\big[\phi(S) \mid \tilde{D}_M\big] - \mathbb{E}\big[\phi(S) \mid \tilde{D}_N\big]\big)^2$$
$$+ 2\left(\phi(S)\mathbb{E}\big[\phi(S) \mid \tilde{D}_M\big] - \phi(S)\mathbb{E}\big[\phi(S) \mid \tilde{D}_N\big] - \mathbb{E}\big[\phi(S) \mid \tilde{D}_M\big]^2 + \mathbb{E}\big[\phi(S) \mid \tilde{D}_M\big]\mathbb{E}\big[\phi(S) \mid \tilde{D}_N\big]\right)\Big]$$

$$=\mathbb{E}_{\tilde{D}_N,\phi(S),\tilde{D}_M}\left[\big(\phi(S) - \mathbb{E}\big[\phi(S) \mid \tilde{D}_M\big]\big)^2 + \big(\mathbb{E}\big[\phi(S) \mid \tilde{D}_M\big] - \mathbb{E}\big[\phi(S) \mid \tilde{D}_N\big]\big)^2\right]$$
$$+ 2\left(\mathbb{E}_{\tilde{D}_N,\phi(S),\tilde{D}_M}\left[\phi(S)\mathbb{E}\big[\phi(S) \mid \tilde{D}_M\big]\right] - E_{\tilde{D}_N,\phi(S),\tilde{D}_M}\left[\phi(S)\mathbb{E}\big[\phi(S) \mid \tilde{D}_N\big]\right]\right.$$
$$\left. - \mathbb{E}_{\tilde{D}_N,\phi(S),\tilde{D}_M}\left[\mathbb{E}\big[\phi(S) \mid \tilde{D}_M\big]^2\right] + \mathbb{E}_{\tilde{D}_N,\phi(S),\tilde{D}_M}\left[\mathbb{E}\big[\phi(S) \mid \tilde{D}_M\big]\mathbb{E}\big[\phi(S) \mid \tilde{D}_N\big]\right]\right)$$

$$=\mathbb{E}_{\tilde{D}_N,\phi(S),\tilde{D}_M}\left[\big(\phi(S) - \mathbb{E}\big[\phi(S) \mid \tilde{D}_M\big]\big)^2 + \big(\mathbb{E}\big[\phi(S) \mid \tilde{D}_M\big] - \mathbb{E}\big[\phi(S) \mid \tilde{D}_N\big]\big)^2\right]$$
$$+ 2\left(\mathbb{E}_{\tilde{D}_N,\tilde{D}_M}\mathbb{E}_{\phi(S)|\tilde{D}_N,\tilde{D}_M}\left[\phi(S)\mathbb{E}\big[\phi(S) \mid \tilde{D}_M\big]\right] - \mathbb{E}_{\tilde{D}_N,\tilde{D}_M}\mathbb{E}_{\phi(S)|\tilde{D}_N,\tilde{D}_M}\left[\phi(S)\mathbb{E}\big[\phi(S) \mid \tilde{D}_N\big]\right]\right.$$
$$\left. - \mathbb{E}_{\tilde{D}_N,\phi(S),\tilde{D}_M}\left[\mathbb{E}\big[\phi(S) \mid \tilde{D}_M\big]^2\right] + \mathbb{E}_{\tilde{D}_N,\phi(S),\tilde{D}_M}\left[\mathbb{E}\big[\phi(S) \mid \tilde{D}_M\big]\mathbb{E}\big[\phi(S) \mid \tilde{D}_N\big]\right]\right)$$

$$=\mathbb{E}_{\tilde{D}_N,\phi(S),\tilde{D}_M}\left[\big(\phi(S) - \mathbb{E}\big[\phi(S) \mid \tilde{D}_M\big]\big)^2 + \big(\mathbb{E}\big[\phi(S) \mid \tilde{D}_M\big] - \mathbb{E}\big[\phi(S) \mid \tilde{D}_N\big]\big)^2\right]$$
$$+ 2\left(\mathbb{E}_{\tilde{D}_N,\tilde{D}_M}\mathbb{E}_{\phi(S)|\tilde{D}_M}\left[\phi(S)\mathbb{E}\big[\phi(S) \mid \tilde{D}_M\big]\right] - \mathbb{E}_{\tilde{D}_N,\tilde{D}_M}\mathbb{E}_{\phi(S)|\tilde{D}_M}\left[\phi(S)\mathbb{E}\big[\phi(S) \mid \tilde{D}_N\big]\right]\right.$$
$$\left. - \mathbb{E}_{\tilde{D}_N,\phi(S),\tilde{D}_M}\left[\mathbb{E}\big[\phi(S) \mid \tilde{D}_M\big]^2\right] + \mathbb{E}_{\tilde{D}_N,\phi(S),\tilde{D}_M}\left[\mathbb{E}\big[\phi(S) \mid \tilde{D}_M\big]\mathbb{E}\big[\phi(S) \mid \tilde{D}_N\big]\right]\right)$$

$$=\mathbb{E}_{\tilde{D}_N,\phi(S),\tilde{D}_M}\left[\big(\phi(S) - \mathbb{E}\big[\phi(S) \mid \tilde{D}_M\big]\big)^2 + \big(\mathbb{E}\big[\phi(S) \mid \tilde{D}_M\big] - \mathbb{E}\big[\phi(S) \mid \tilde{D}_N\big]\big)^2\right]$$
$$+ 2\left(\mathbb{E}_{\tilde{D}_N,\tilde{D}_M}\left[\mathbb{E}\big[\phi(S) \mid \tilde{D}_M\big]\mathbb{E}\big[\phi(S) \mid \tilde{D}_M\big]\right] - \mathbb{E}_{\tilde{D}_N,\tilde{D}_M}\left[\mathbb{E}\big[\phi(S) \mid \tilde{D}_M\big]\mathbb{E}\big[\phi(S) \mid \tilde{D}_N\big]\right]\right.$$
$$\left. - \mathbb{E}_{\tilde{D}_N,\phi(S),\tilde{D}_M}\left[\mathbb{E}\big[\phi(S) \mid \tilde{D}_M\big]^2\right] + \mathbb{E}_{\tilde{D}_N,\phi(S),\tilde{D}_M}\left[\mathbb{E}\big[\phi(S) \mid \tilde{D}_M\big]\mathbb{E}\big[\phi(S) \mid \tilde{D}_N\big]\right]\right)$$

$$=\mathbb{E}_{\tilde{D}_N,\phi(S),\tilde{D}_M}\left[\big(\phi(S) - \mathbb{E}\big[\phi(S) \mid \tilde{D}_M\big]\big)^2 + \big(\mathbb{E}\big[\phi(S) \mid \tilde{D}_M\big] - \mathbb{E}\big[\phi(S) \mid \tilde{D}_N\big]\big)^2\right]$$

Assume a measure space exists such that $\lim_{M \to \infty} \tilde{D}_M$ is well defined. Then we can alternatively replace $\tilde{D}_M$ with $\tilde{D}_\infty = \lim_{M \to \infty}$ in the above decomposition and consider $\mathbb{E}\big[\tilde{\phi}(S) \mid \tilde{D}_\infty\big]$ the optimal predictor of $\phi(S)$ given an infinite dataset. Combining all terms, we arrive at the desired decomposition:

$$\mathbb{E}\left[\big(\phi(S) - f_\theta(\tilde{D})\big)^2\right] = \underbrace{\mathbb{E}\left[\big(\phi(S) - \tilde{\phi}(S)\big)^2\right]}_{\substack{\text{A. MC error in the} \\ \text{targeted causal parameter}}} + \underbrace{\mathbb{E}\left[\big(f_\theta(\tilde{D}_N) - \mathbb{E}\big[\phi(S) \mid \tilde{D}_N\big]\big)^2\right]}_{\substack{\text{B. model error from the} \\ \text{optimal predictor given } N \text{ samples}}}$$

$$+ \underbrace{\mathbb{E}\left[\big(\phi(S) - \mathbb{E}\big[\phi(S) \mid \tilde{D}_\infty\big]\big)^2\right]}_{\substack{\text{C. identification error (uncertainty} \\ \text{in the causal estimand given the best} \\ \text{predictor with an infinite dataset)}}} + \underbrace{\mathbb{E}\left[\big(\mathbb{E}\big[\phi(S) \mid \tilde{D}_\infty\big] - \mathbb{E}\big[\phi(S) \mid \tilde{D}_N\big]\big)^2\right]}_{\substack{\text{D. finite sample error} \\ \text{of the optimal predictor}}}$$

$\square$

## B    Baseline identification conditions

Because BBCI supports causal inference with varying identification assumptions, we detail the identification assumptions for a few common settings here that justify in each case the use of an MLP during estimation, in correspondence with the per-dataset regression baselines used in Section 3. Recalling notation from the main text, let $\tilde{y}, \tilde{t}, \tilde{x}, \tilde{u}_t, \tilde{w}$ denote the outcome, treatment, confounders, instrument variables, and proxies respectively.

### B.1    Two-stage instrument variable methods

Instrumental variables (IVs) $\tilde{u}_t$ only affect the outcome $\tilde{y}$ through the treatment assignment $\tilde{t}$ and do so independently of confounders $\tilde{x}$. This allows us to isolate changes in $\tilde{y}$ that are due only to treatment. Traditional IV methods perform a two-stage least-squares (TSLS) procedure that assumes linear and homogeneous treatment effects, modeling as linear both the effect of the treatment on the outcome and the effect of the instrument on the treatment. Following Puli & Ranganath (2020), assuming that the outcome generating process is linear,

$$\tilde{y} = \beta_t \tilde{t} + \beta_x \tilde{x} + \texttt{zero-mean noise},$$

identification in TSLS comes from having 1) an IV $\tilde{u}_t$ that is marginally independent of the confounder, and 2) the conditional expectation $\mathbb{E}[\tilde{t} \mid \tilde{u}_t]$ not be a constant function of the IV. In this case, the causal effect is identified as the linear coefficient of $\mathbb{E}[\tilde{t} \mid \tilde{u}_t]$ when regressing $\tilde{y}$ on $\mathbb{E}[\tilde{t} \mid \tilde{u}_t]$, because

$$\mathbb{E}[\tilde{y} \mid \tilde{u}_t] = \beta_t \mathbb{E}[\tilde{t} \mid \tilde{u}_t] + \mathbb{E}[\tilde{x} \mid \tilde{u}_t],$$

where $\mathbb{E}[\tilde{x} \mid \tilde{u}_t] = \mathbb{E}[\tilde{x}]$ is a constant and $\mathbb{E}[\tilde{t} \mid \tilde{u}_t]$ varies with $\tilde{u}_t$ due to assumptions 1 and 2, respectively.

This identification result allows for non-parametric estimation of $\mathbb{E}[\tilde{t} \mid \tilde{u}_t]$, for which we use an MLP.

### B.2    Regression-based proximal causal inference

Liu et al. (2024) prove identification for a setting with proxy variables (instead of IVs in the TSLS case). The idea is, like in TSLS, exploit the linearity of the outcome generation process. Given a treatment-side proxy $\tilde{w}_1$ and outcome-side proxy $\tilde{w}_2$ such that exclusion holds $\tilde{w}_1 \perp (\tilde{t}, \tilde{w}_2) \mid \tilde{x}$, and $\tilde{w}_2 \perp (\tilde{y}, \tilde{w}_2) \mid \tilde{x}, \tilde{t}$, Liu et al. (2024) consider the case where the proxy and the outcome generation processes follow:

$$\mathbb{E}[\tilde{w}_1 \mid \tilde{t}, \tilde{w}_2] = \alpha_1 + \alpha_g \mathbb{E}[\tilde{x} \mid \tilde{t}, \tilde{w}_2]$$

$$\mathbb{E}[\tilde{y} \mid \tilde{t}, \tilde{w}_2] = \beta_y + \beta_t \tilde{t} + \mathbb{E}[\tilde{x} \mid \tilde{t}, \tilde{w}_2],$$

where $\alpha_g \neq 0$ to make the $\tilde{w}_1$ an informative proxy of $\tilde{x}$. One can re-arrange terms and write

$$\mathbb{E}[\tilde{y} \mid \tilde{t}, \tilde{w}_2] = \texttt{constant} + \beta_t \tilde{t} + \frac{1}{\alpha_g} \mathbb{E}[\tilde{w}_1 \mid \tilde{t}, \tilde{w}_2].$$

Thus, similar to the case of TSLS, $\beta_t$ is identified as the coefficient of $\tilde{t}$ in the regression of $\tilde{y}$ on $\tilde{t}$ and $\mathbb{E}[\tilde{w}_1 \mid \tilde{t}, \tilde{w}_2]$. As in the TSLS case in Appendix B.1, this identification result allows for non-parametric estimation of $\mathbb{E}[\tilde{w}_1 \mid \tilde{t}, \tilde{w}_2]$, for which we use an MLP.

## C    Additional experiments

### C.1    Higher-dimensional covariates

We use the following adaptation of DGP (2) to additionally test that BBCI can accommodate higher-dimensional covariates. In this section, we focus on the confounder setting and $N = 100$ as a demonstration. The modification of DGP (2) instead uses $d$-dimensional confounder, instrument, and outcome noise vectors $\tilde{x}^{(i)}, \tilde{u}_t^{(i)}, \tilde{u}_y^{(i)} \sim \mathcal{N}(\mathbf{0} \in \mathbb{R}^d, I_{d \times d})$ with corresponding $d$-dimensional vectors of coefficients $\boldsymbol{\gamma_x}, \boldsymbol{\beta_x}, \boldsymbol{\gamma_t}, \boldsymbol{\beta_y} \in \mathbb{R}^d$, now with inner products instead of scalar multiplication. Each coordinate of $\boldsymbol{\gamma_x}, \boldsymbol{\beta_x}, \boldsymbol{\gamma_t}, \boldsymbol{\beta_y}$ is sampled in

the same manner as the corresponding $\gamma_x, \beta_x, \gamma_t, \beta_y$ in the original scalar case. In this case, the embedding size was increased to 256 and number of encoding layers was increased to 12, trained again for 200 epochs. Results in Table 6 show that BBCI is indeed not limited to the scalar case; however, higher-dimensional covariates require more computation. Note also that the IHDP experiment in Section 3.6 has 25-dimensional covariates and the LaLonde experiment in Section 5 has 9-dimensional covariates.

Table 6: PATE estimation for the confounder setting of DGP (2) with five-dimensional covariates.

| Setting | Model | $R^2$ | RMSE |
|---|---|---|---|
| Confounder, $N = 100$ | T-Only-Transformer | 0.7152 | 0.3059 |
| Confounder, $N = 100$ | BBCI | **0.9531** | **0.1242** |
| Confounder, $N = 100$ | Reg-MLP | 0.8999 | 0.1814 |
| Confounder, $N = 100$ | Reg-Lin | 0.9464 | 0.1327 |

## D BBCI computational requirements and efficiency

Table 7: Details on the number of epochs, number of gradient steps, and running time of BBCI.

| Experiment ($N = 1000$) | Number of epochs | Number of gradient steps | Running time |
|---|---|---|---|
| Linear PATE | 30 | 4710 | < 15 minutes |
| CATE | 30 | 4710 | < 15 minutes |
| Nonlinear PATE, MLPs | 200 | 31400 | 9 hours |
| Nonlinear PATE, Trees | 200 | 31400 | 11 hours |
| Nonlinear PATE, Splines | 200 | 31400 | 17 hours |
| IHDP | 100 | 15700 | 3 hours |
| LaLonde | 100 | 23400 | 3 hours |

Table 7 shows details about the number of epochs, number of gradient steps, and running time across several of the above experiments to characterize the running time of BBCI. BBCI, as one might expect from a meta-prediction task, takes more time to train than a corresponding prediction task. However, once trained, a new prediction task requires only a forward pass of the model.

