# OpenReview forum: "Black Box Causal Inference: Effect Estimation via Meta Prediction"
_TMLR — Rejected by TMLR_

### Review · Reviewer_n25L · 2025-04-20

**Summary Of Contributions:**

This paper studies causal effect estimation problem in a black-box manner, by learning to estimate the effects.
The goal is bypassing the complexities of choosing and/or designing the appropriate algorithms for different settings. As such, BBCI (black-box causal inference) is aimed to work with different identification assumptions and data settings.

The approach is based on simulating dataset-effect pairs from samples of an SCM belonging to a family of SCMs, and learning to predict the effect from data. BBCI consists of the steps: choosing an SCM family $\cal F$ and model class $f_{\theta}$ (chosen as set transformer ++ framework); then at each training step, sampling an SCM $S$ from the family and sampling a dataset for $S$, computing the target estimand for $S$, and updating model parameters $f_{\theta}$ in a supervised fashion.

To demonstrate the effectiveness of the proposed approach, the paper compares BBCI’s performance to baseline methods under different settings with known identification settings and some real datasets. Experiment results are promising as BBCI outperforms (or match) the performance of the competitors,

**Audience:**

Yes

**Broader Impact Concerns:**

N.A

**Claims And Evidence:**

Yes

**Requested Changes:**

- Theorem 1: Training error for BBCI is broken down to four parts, each of which makes sense. However, there is no experiment or further discussion on the interplay between these parts. For instance, for the different settings discussed in Sections 3, 4, and 5, can you discuss how each of these components are expected to differ? Some implications are obvious, though as a reader, I’d just want to see a more thorough treatment.
- Parameterization of data generation processs (DGP) eq. (2): According to Table 1, BBCI outperforms the competitors very clearly. I wonder what are some possible explanations for the huge difference in performance, and whether parameterization of DGP creates a bias to either side (in favor of BBCI or the competitors), since only one DGP is considered in Table 1.

Very minor points:
- References: It’d be good to revise the bibliography. I’d suggest citing the most recent versions of the papers (e.g., citing the published versions instead of the preprint ones, I see at least 3 such cases).
- Typos: ``targted’’ in page 1. Using both 1970s,1980s  and 50s, 70s, I just suggest being consistent.

**Strengths And Weaknesses:**

- The learning based approaches (especially using simulated data) for causal inference problems are growing in recent years. This paper represents a good effort in one aspect of such approaches.
- The claims are backed up by experiments. Especially known SCM class setting (Section 3) shows strong results under multiple relevant cases, e.g., IV variables and mixed datasets.
- There are also experiments on real datasets — this is important as the success of simulation-based-training is not guaranteed to transfer to real data.
- That being said, the experiments on real data and $\cal F \neq \cal F_0$ (Section 4) cases are rather limited, and would benefit from additional demonstration. Also, parameterization of Section 3 experiments can benefit from more discussion.
- The implications/interpretation of Theorem 1 is not very clear.

---

> ### Author Response · Authors · 2025-05-05
>
> Thank you for the review, comments, and additional suggestions. Based on the requested changes, we have run additional experiments to further explore and demonstrate the different terms in the error decomposition in Theorem 1 as well as the role of different parameters in baseline performance for Table 1.
>
> Please see [Figure 1 at the following link](https://anonymous.4open.science/api/repo/bbci_tmlr/file/tmlr_rebuttal.pdf?v=c6738206) for the first point about Theorem 1. This figure shows the result of three experiments to demonstrate the role of different terms in the error decomposition. Each point in each plot corresponds to a BBCI model of the same size (4-layer ST++ with embedding size 128) trained for the same number of epochs (200) using the DGP in Equation 2. In each experiment, BBCI is trained using approximate target PATE $\tilde{\phi}$, but plots show the test $R^2$ computed with respect to the true exact PATE $\phi$. Panel (a) shows the role of increasing the number of Monte Carlo (MC) samples used to estimate the true parameter $\phi$ during training. As we use a higher number of MC samples, error decreases ($R^2$ increases), corresponding to decreasing term A in the error decomposition. Panel (b) shows the same trend if we increase dataset size, corresponding to decreasing finite sample error in term D. Panel (c) introduces a new variable into Equation 2 — a confounder $x_h$ that is always hidden and contributes to treatment $t$ and outcome $y$ with the same coefficients as observable confounder $x$, now scaled by constant $\alpha$. In other words, we add the following variable $x_h$ and modify the equations for treatment $t$ and outcome $y$ as follows:
>
> $$x_h \sim \mathcal{N}(0, 1);$$
> $$t = \gamma_x x + \alpha \gamma_x x_h + \gamma_t u_t;$$
> $$y = \beta_x x + \alpha \beta_x x_h + \beta_t t + \beta_y u_y$$
>
> As $\alpha$ is varied, the strength of hidden confounder $x_h$ is scaled up or down. When $\alpha=1$, $x_h$ shares the same coefficients with $x$. Comparing performance in the observed confounding setting, where $x$ is observed, to performance in the IV setting, where $u_t$ is observed, demonstrates the role of identification error in the error decomposition (term C). With nonzero hidden confounder $x_h$, identification is no longer satisfied in the observed confounder setting, and error increases. However, in the IV setting, identification is unaffected by $x_h$, and error correspondingly does not exhibit the same decrease. The remaining model error term in the decomposition (term B) we would expect to decrease with more training, but as error decreases across all cases with more training, we do not show a direct visualization for this term. Together, these results help provide additional illustration and intuition for the terms in Theorem 1.
>
> Regarding the second point about further analysis of the differences in performance between BBCI and baselines in Table 1, one of the main drivers of poor baseline performance we were able to discover is the amount of noise on the treatment assignment, controlled by parameter $\gamma_t$. Low values of  $\gamma_t$ correspond both to having a weak instrument $u_t$ as well as to having stronger confounding from confounder $x$, which affects all baselines. Absolute errors for 1000 predictions in each setting are plotted against parameter magnitudes in Figure 2a at the same link. The column for $\gamma_t$ shows this is the most visible effect across baselines. But apart from the most visible effects from $\gamma_t$, other parameters also impact baseline algorithm performance. This includes, in the proxy case and especially in the confounder case, how much noise there is on the outcome (parameterized by $\beta_y$). Additionally in the proxy case, we see how strong the proxies are plays a role, where higher variances $\delta_1$ and $\delta_2$ lead to worse performance of proximal 2SLS method. To highlight these other effects, which are somewhat visible in 2a but less impactful than $\gamma_t$, Figure 2b at the same link shows the same plot with a restricted range for $\gamma_t$ to be above 0.2, making trends wrt the other parameters more visible. While the focus of our work is not on the specific failure modes of the different baseline algorithms, we agree with this question from the reviewer and will add these summaries to the appendix to further contextualize the results of Table 1. Each possible baseline algorithm that performs worse than BBCI likely suffers from some failure modes that could be separately analyzed in order to draw direct comparisons between a given learned estimation algorithm from BBCI and a setting-specific estimation algorithm. We would describe doing an even deeper such analysis as “explaining” the estimation methods learned by BBCI, which we view as an important direction for additional future work.
>
> Thank you also for the minor points about references and typos, we will be sure to fix those.

---

### Review · Reviewer_Mebz · 2025-04-27

**Summary Of Contributions:**

The proposed method, black box causal inference (BBCI), aims to “learn to estimate causal effects in any setting where identification holds.” That is, given data $D := \\{(x, y, t)_i \\} _{i<N} \sim P(x, y, t)$ with *known* SCMs and *ground-truth* ATE or CATE $q_D$, BBCI aims to learn a mapping from $D$ to $q_D$ which could hopefully be applied to new datasets $D'$ with *unknown* SCMs. In this paper, the method learns from $(D, q_D)$ pairs *simulated from a single causal graph*, though some experiments consider different outcome functions. Three causal graphs are considered: unconfounded, IV, and two-proxies.

**Audience:**

Yes

**Claims And Evidence:**

No

**Requested Changes:**

I am afraid that, if I have not made significant mistakes in my above evaluation, the project needs a thorough rework. The authors are welcome to correct me if I am wrong, or use the above points as suggestions for future work.

An immediate next step would be to show that BBCI works for CATE and ATE (with *unknown* propensity!) under unconfoundedness.

**Strengths And Weaknesses:**

### Strengths

The problem setting of the paper is interesting and ambitious, and resembles a “foundation model for causal inference.”

### Weaknesses

The high-level designs of the method and experiments are not convincing. Below I explicate these points together with my own thoughts, and hope they will be helpful.

1. The authors are right to point out the long-standing endeavor in causal inference. But why is it so hard even for humans? First, we need to determine under which SCMs the target query is identifiable and what the identification is, i.e., how to write the query as a functional of the observational distribution. If we know that identifiability holds, as required by the method, then we have solved at least half of the problem! Otherwise, when *we do not know whether the data-generating SCM gives identifiable queries,* **there is no guarantee the proposed method will work**, either.

2. Another half of the difficulty is, as the authors say, to design valid estimators based on the identification. But why? Because, importantly, we need both structural and parametric conditions from the identification. We need first to exploit **structural knowledge**, or else, we don’t even know what observational distributions we could use as building blocks for identification equations (IDEQs). For example, under IV, the IDEQ involves $P(T|I)$; intuitively, we want to separate $T$ into two parts, explained by $I$ (the IV) or not. But we are led to this by knowing $I$ affects $Y$ only through $T$. Thus, for automatic designs of valid estimators, it is essential to learn to exploit structural information. However, *in this method, structural info is manually built-in* (via $\phi$ in Algo 1, which is fixed but not learned), so it cannot automatically adapt “in any setting,” contrary to the stated goal.

3. Specifically, $\phi$ needs to condition on all simulated endogenous variables directly affecting $Y$. This point seems never clearly explained in the paper. Also, it is seriously confusing to overload the notation $X$ here, because *for some graphs, conditioning on some of the covariates $X$ might be wrong*, e.g., when one of $x$ is a common effect of $t$ and $y$, or the “M-bias” case. Ultimately, **$\phi$ needs to be tailored for different graphs.** In addition, the symbol $\phi$ is also confusing here, because at the beginning of the paper, $\phi$ refers to a learnable mapping from $D$ to $q_D$, but now $\phi$ is (structurally) fixed while $f$ is the learnable mapping.

4. Then, based on structural knowledge, we need to exploit *parametric, but non-distributional* information. Indeed, for IV and two-proxy settings, we cannot solve the integral IDEQs without the parametric conditions, e.g., monotonicity and completeness, both of which are *non-testable*! That is, even if knowing we have, say, the IV causal structure, *we cannot learn, from observational distributions $\\{P\\}$, whether $P$ is generated from an identifiable SCM*, *no matter how many $P$’s we simulate*, because there can be an identifiable setting and an unidentifiable one, which generate the same $P$.

5. Now, to sum up, it is problematic to conceptualize the framework as learning a map from $D$ to $q_D$. Rather, *the map we want to learn should take in “all info needed for identification”*, some conditions like completeness cannot even be fully described by SCMs. By far, I have talked about identification, and hope I have made it clear we cannot bypass it and jump directly to estimation. Certainly, estimation is also hard! Now the focus is instead on **finite sample performance**, with problems like balance under unconfoundedness and bias amplification under IVs. However, *the method glosses over this by sampling as much data as possible from the sampler with **known SCMs**.*

6. The experimental settings considered in the paper are not particularly useful.
    1. Setting 1 (explained in Sec 1.1 & experimented in Sec 3), where most experiments are done. Here, **the full identifiable SCMs are known** (as the same in the data simulator/sampler). There are no problems to solve here except finite sample performance.
    2. Setting 2 (explained in Sec 1.2 & experimented in Sec 4), where $p(x)$ and $p(t|x)$ are known. CATE is not considered for this setting, and the experiment examines *ATE under unconfoundedness, for which, **knowing propensity** $p(t|x)$ means any outcome regression on $t$, after adjustment by the propensity, will work* (this is the idea of IPW). Again, there are no problems to solve here. Moreover, it is unclear between which two variables the $R^2$ is computed here and throughout the experiments.
    3. Setting 3 (explained in Sec 1.3 & experimented in Sec 5). **The LaLonde data is nearly unconfounded**, which can be seen from the performance of other methods. Then, by building the unconfoundedness into BBCI, the problem is nearly solved. The remaining gain, I believe, can be explained by the better fitting capacity of the transformer. As a sanity check, it would be useful to test the method, which is trained on the unconfounded setting, instead on the IV setting. Another concern is the poor performance under $N=100$; I suspect this is due to the modeling of $p(x)$ from data. Which model did you use for $p(x)$? With small data, the fitting of $p(x)$ might not be good.

7. Finally, in the current form, the method only uses datasets containing a single point in the space of causal graphs, i.e., a single causal graph. We would like to learn from thousands and millions of causal graphs, to achieve the claimed ambitious goal of “learn[ing] to estimate causal effects in any setting where identification holds.”

Clarity is also a problem in the paper, but I will not go into detail here because I have spent much time sorting out the above points, which I hope contribute to a useful review.

---

> ### Author Response · Authors · 2025-05-05
>
> Thank you for the review and comments. There seems to be a key misunderstanding/ misconception we wish to clarify before further discussing individual points in the review.
>
> # Main misconception — one meta-algorithm, not one meta-model
> **We do not train a single model to simultaneously cover all identification conditions. Rather, we show how the same *algorithm* can be used to train a model for each identification condition separately.** We do not claim to solve the problem of identification, but instead focus on learning to estimate once identification is given. Our decomposition in Theorem 1 explicitly shows that this approach is *not* a silver bullet, and that if the quantity of interest is not identified, the approach incurs error. Furthermore, it directly acknowledges the challenge of using synthetic data. The line before our problem statement summarizes our focus on **simplifying estimator development**, rather than on learning a single mapping that covers all identification conditions: “We propose to use meta-learning to simplify estimator development, both in settings with known estimators and in settings without well-established estimators, leading us to the following problem statement.”
>
> We will further clarify this point in the paper both in writing and with an explicit demonstration of the following failure case:
> - Consider when the target quantity is not identifiable. For example, consider **[Panel C in Figure 1 at the following link](https://anonymous.4open.science/api/repo/bbci_tmlr/file/tmlr_rebuttal.pdf?v=c6738206)**. In such a case, BBCI training error will be stuck at a low value, and won't simply give us misleading results. This corresponds to term C for ‘identification error’ in the error decomposition in Theorem 1, where uncertainty in the causal estimand can remain even given the best predictor with an infinite dataset. Detecting such issues does require having sufficiently large datasets for each sample during training (compared to, e.g., predicting on a single test point), which we will also further acknowledge in the discussion.
>
> Because we train a separate BBCI regressor for each identifiable problem, **there is no reason for us to explicitly provide the conditions that make the problem identifiable to BBCI**. However, this would definitely be necessary for extending BBCI to a "Meta-BBCI" approach where one regressor works for different target problems, and this extension suggested by the reviewer would be a great future direction.
>
> The above general clarification addresses the following points from Weaknesses 1, 2, 3, 4, 5, 6, and 7. Where applicable, we elaborate further on the general clarification:
> - (W1) "When we do not know whether the data-generating SCM gives identifiable queries, then there is no guarantee the proposed method will work, either"
>     - This point is in line with the above discussion of Theorem 1 including identification error.
> - (W2 & W3) "in this method, structural info is manually built-in [...] so it cannot automatically adapt ‘in any setting,’ contrary to the stated goal." & "$\phi$ needs to be tailored for different graphs"
>     - We can clarify as above that our stated goal is not for one trained model to adapt to any setting, and that we do in fact tailor $\phi$ for each graph when training each model.
> - (W4) "we cannot learn, from observational distributions $\{P\}$, whether $P$ is generated from an identifiable SCM, no matter how many $P$’s we simulate"
>     - The above clarifies that we are not making this claim.
> - (W5) "the map we want to learn should take in 'all info needed for identification'" / "Now the focus is instead on finite sample performance [...] However, the method bypasses this by sampling as much data as it can from the sampler with known SCMs."
>     - We do not bypass finite sample performance, but instead sample only as much data as is available in whatever source dataset or setting on which we want to run estimation. Finite sample error plays an explicit role as term D in Theorem 1.
> - (W6.3) "As a sanity check, it would be useful to test the method, which is trained on the unconfounded setting, instead on the IV setting."
> - (W7) "We would like to learn from thousands and millions of causal graphs, to achieve the claimed ambitious goal of 'learn[ing] to estimate causal effects in any setting where identification holds.'"
>     - As we discussed above, BBCI builds models for each identification setting separately. To use BBCI for a setting therefore does not require sampling from millions of causal graphs.

---

> > ### Author Response · Authors · 2025-05-05
> >
> > # Additional responses
> > We next address a few additional separate points that were mentioned across the weaknesses:
> > - (W3) "it is seriously confusing to overload the notation $X$ here, because for some graphs, conditioning on some of the covariates $X$ might be wrong"
> >     - We have used the notation such as $p(t | x)$ and $p(y | x,t)$ to describe the DGPs rather than to describe which covariates to condition on for identification of the effect. While $p(t | x)$ and $p(y | x,t)$ correctly describe all the cases we consider in the paper, we agree with the reviewer that there are many cases like colliders and M-bias where we would not condition on all covariates in order to recover the effect. We can clarify this point as well as the notation.
> > - (W6.1) "[In Setting 1], the full identifiable SCMs are known (as the same in the data simulator/sampler). There are no problems to solve here except finite sample performance."
> >     - We believe this setting is useful for the development of efficient estimation algorithms for settings with known identification, which otherwise can require significant manual work or even decades of research (e.g., the many years it took to go from regression-based methods and IPW to AIPW and TMLE for the observed confounding case, and the similarly long trajectory for improving estimation methods in the IV case). Additionally, some settings, even though identification is known, do not yet have efficient or established estimation algorithms.
> > - (W6.2) "knowing propensity $p(t|x)$ means any outcome regression on $t$, after adjustment by the propensity, will work (this is the idea of IPW). Again, there are no problems to solve here. Moreover, it is unclear between which two variables the $R^2$ is computed here and throughout the experiments."
> >     - The purpose of Setting 2 is to explore generalization across SCM families, not to say that there are not existing methods (such as IPW) for estimation with known propensity $p(t|x)$. Further, BBCI is not given knowledge of $p(t \mid x)$.
> > The reported $R^2$ is between predicted and true PATE values.
> > - (W6.3) "The LaLonde data is nearly unconfounded, which can be seen from the performance of other methods"
> >     - In this case, we introduce confounding into the LaLonde data by adding observational controls, as in [Dehejia & Wahba 1999]. The presence of confounding is evidenced by the sign difference when estimating using a difference in means ("TreatmentOnly") compared to the randomized estimate ("Randomized"). The other methods control for confounding, so although their performance is better, that does not indicate that the data is unconfounded.
> > - (W6.3) "Which model did you use for $p(x)$? With small data, the fitting of $p(x)$ might not be good."
> >     - BBCI does not model $p(x)$; the data used to train BBCI is produced by using the observed data for $x$, i.e., we use the empirical distribution and simulate outcomes.

---

> ### Comment · Reviewer_Mebz · 2025-05-19
> **Main concerns remain**
>
> “Meta-algorithm”
>
> This seems like a mild conceptual or wording tweak. I referred to your method as resembling a “foundation model for causal inference” because you described it as “a method to learn to estimate causal effects in any setting where identification holds,” that works “in settings without well-established estimators,” and focuses on “simplifying estimator development.” My main concern is, and has been, that your method — whatever it is called — cannot achieve these goals, as I argued, without incorporating knowledge about identification assumptions.
>
> If you insist on this possibility, you could try to **experiment with the method in a setting that is identifiable but lacks existing estimators** — which, as I indicated in my original review, is difficult to even find (for good reasons). Or suppose, I’m a practitioner with some data and some domain knowledge. If I don't know the identifiability, which is more often the case, how would you engage with me to determine whether your method applies to my problem? If I know the identification, how would you convince me to use your method, instead of developing my own? Stressing it is hard is not useful; it is hard because we want to address finite-sample issues, which your method doesn't address.
>
> Returning to my original points:
>
> Structural knowledge.
> If your method needs to be *tailored to each setting based on the graph*, even given the identification, then in what sense does it “simplify estimator development”?
>
> Please note that It is not a contribution to just *omit finite-sample issues* like *imbalance* under unconfoundedness or *bias amplification* under IVs. You need to at least **show experimentally that your method performs as well as previous sample-efficient methods under limited data**.
>
> W4 (Non-Structural knowledge).
> My point here was exactly that your method does not use the non-structural information that is necessary for identification — in other words, it treats an identifiable setting as if it were unidentifiable.
>
> W5.
> Your “source dataset” comes from a simulator with known SCMs, and you *tailored your method based on these known SCMs*.
>
>
>
> Other points:
>
> In Sections 2.1 and 2.2, you do use 𝑋 to describe which covariates to condition on for identifying the effect.
>
> Setting 1: I want to reiterate that estimation is hard because of inite-sample issues like **imbalance** under unconfoundedness or bias amplification under IVs. The setting you present is trivial if not for addressing these issues.
>
> Setting 2: Since the true propensity is used to generate the training data, it is effectively known. My point wasn’t that estimators  (IPW) exist, but that the setting is trivial unless you are **estimating CATE**, which you are not.
>
> “The reported R² is between predicted and true PATE values.” I don’t understand the motivation for reporting this. R² measures linear fit; why should we expect predicted and true PATE values to be linearly related (except that they should be as close as possible)? On the other hand, even if they are linearly related, the error could be arbitrarily large.
>
> Regarding the LaLonde data, my point was that hidden confounding, if there is any, is very weak — as evidenced by the strong performance of methods that only control for observed confounding.
>
> **Bold** face contains a minimal action list for the authors in case they believe they are doable.

---

> > ### Author Response · Authors · 2025-05-26
> >
> > Thank you for the response about remaining concerns. We respond to each of the main followup points below.
> >
> > > “Meta-algorithm” This seems like a mild conceptual or wording tweak.  I referred to your method as resembling a “foundation model for causal inference” because you described it as “a method to learn to estimate causal effects in any setting where identification holds,” that works “in settings without well-established estimators,” and focuses on “simplifying estimator development.” My main concern is, and has been, that your method — whatever it is called — cannot achieve these goals, as I argued, without incorporating knowledge about identification assumptions.”
> >
> > - This is more than a mild wording tweak, but an important distinction: rather than train a single “foundation model” to simultaneously cover all identification conditions, we use the same procedure/algorithm to train many different models for many different identification conditions, separately.
> > - We would like to reiterate that **there is no reason for us to explicitly provide the conditions that make the problem identifiable to BBCI, which learns the adjustment in a data-driven way**.
> > - This point also underlines an important distinction. When we make use of identification information in order to simulate data (such as propensity scores, as you mention) this is in order to evaluate the performance of BBCI with known ground truth data.
> >
> > > If you insist on this possibility, you could try to experiment with the method in a setting that is identifiable but lacks existing estimators — which, as I indicated in my original review, is difficult to even find (for good reasons).
> >
> > - **We show experimental results for a setting that is identifiable but lacks existing estimators in Table 5**.
> >
> > >You need to at least show experimentally that your method performs as well as previous sample-efficient methods under limited data.
> >
> > - **We show experimentally that BBCI performs as well as double machine learning methods in Table 2**.
> >
> > >Setting 1: I want to reiterate that estimation is hard because of finite-sample issues like imbalance under unconfoundedness or bias amplification under IVs. The setting you present is trivial if not for addressing these issues.
> >
> > - The **performance of regression (Reg-Lin, Reg-MLP), two-stage least squares (2SLS-Lin, 2SLS-MLP), and proximal two-stage least squares (Pr2SLS-Lin, Pr2SLS-MLP) in Table 1 indicates that these settings are not trivial.**
> >
> > >Setting 2: Since the true propensity is used to generate the training data, it is effectively known. My point wasn’t that estimators (IPW) exist, but that the setting is trivial unless you are estimating CATE, which you are not.
> >
> > - The purpose of Setting 2, as we discuss in Section 1.2 and in Section 4, is to demonstrate estimation in cases where the SCM sampler $F$ and the true SCM family $F_0$ are different. The main takeaways here are thus about generalization across SCM families.
> > - As discussed above, **when we make use of identification information in order to simulate data (such as propensity scores) this is in order to evaluate the performance of BBCI with known ground truth data, rather than a claim that we will always know the ground truth.** In this case, we explicitly state that it is only the response surface that differs between $F_0$ and $F$. Thus, in both $F$ and $F_0$ (which are both simulated so that we know the ground truth) the same distribution is used for $p(t \mid x)$.
> >
> > Thank you for brining these points of confusion to our attention. We will include a version of this discussion in the final text to clarify each point.

---

> ### Comment · Reviewer_Mebz · 2025-05-29
> **More concerns arise**
>
> > there is no reason for us to explicitly provide the conditions that make the problem identifiable to BBCI, which learns the adjustment in a data-driven way.
>
> You did not answer my original questions which are deeply related to the **practical usage** of your method: "If I don't know the identifiability, which is more often the case, how would you engage with me to determine whether your method applies to my problem? If I know the identification, how would you convince me to use your method, instead of developing my own?"
>
> > We show experimental results for a setting that is identifiable but lacks existing estimators in Table 5.
>
> The IV setting does *not* require there should be no observed confounders, and the unconfounded setting does not require there should be no IVs, either. So **under your “Confounder+IV” case, either standard estimators for IV or unconfounded settings will work** (Edit: if there are unobserved confounding, standard estimators for unconfoundedness fail, but estimators for IVs still work). It does not make sense to say it "suggest[s] some combination of confounding-specific estimation methods and IV-specific estimation methods..." as if this is a new setting without existing estimators.
>
> > We show experimentally that BBCI performs as well as double machine learning methods in Table 2.
>
> > Table 1 indicates that these settings are not trivial.
>
> As mentioned, your dataset is trivial in the sense that it does not demonstrate finite data issues like **imbalance and bias amplification**, and *any* regression on observed confounders, sample-efficient or not, will work as well. Moreover, **the tables do not show std errors**, so the significance cannot be judged, and I actually believe the differences are insignificant. Also, *why do other methods perform badly outside a range?* This seems like a bug.
>
> > evaluate the performance of BBCI with known ground truth data, rather than a claim that we will always know the ground truth.
>
> This was not my point. But my point is that the response surface, which is the *only* difference between $F_0$ and $F$, actually makes *no* difference for showing "generalization across SCM families." This is because, as mentioned, for ATE under unconfoundedness, *any* outcome regression on $t$, after adjustment by the propensity, will work as well, and the **same** propensity is used for $F_0$ and $F$. **The response surface does make a difference for estimating CATE, but not ATE**, as mentioned.

---

> > ### Author Response · Authors · 2025-05-30
> >
> > >You did not answer my original questions which are deeply related to the practical usage of your method: "If I don't know the identifiability, which is more often the case, how would you engage with me to determine whether your method applies to my problem? If I know the identification, how would you convince me to use your method, instead of developing my own?"
> >
> > We are providing an alternative to a traditional way of designing a new estimator from scratch. In this alternative approach, we focus on a data generating process and leave the estimator design to a meta-learner. We anticipate this alternative would have its own place in addition to (not as a replacement of) the more conventional estimator design.
> >
> > > under your "Confounder+IV" case, either standard estimators for IV or unconfounded settings will work
> >
> > Table 5 shows poor performance for both regression and two-stage least squares in this case. Our experiment also demonstrates that this determination of which estimator to use does not need to be made by a practitioner themself but can be left for our meta-learner to make. We find this a strength, as such determination may not be trivial in other settings.
> >
> > > any regression on observed confounders, sample-efficient or not, will work as well
> >
> > Our results clearly demonstrate empirically that regression does not necessarily work perfectly, especially with a small number of samples. The proposed approach on the other hand works better overall, implying that our meta-learner finds a solution that is effective. We leave it to the future analyzing a particular solution found by our meta-learner; we do agree that the found solution may simply resemble regression.
> >
> > >  the response surface, which is the only difference between $F_0$ and $F$, actually makes no difference for showing "generalization across SCM families" […] The response surface does make a difference for estimating CATE, but not ATE, as mentioned.
> >
> > Figure 4 shows that the response surface does make a difference for estimating ATE and for generalization across SCM families within the meta-learning framework of BBCI — the differences in performance demonstrate that different choices of response surface are not interchangeable.

---

### Review · Reviewer_GQ8V · 2025-05-12

**Summary Of Contributions:**

This paper introduces a novel framework for causal inference, called Black Box Causal Inference (BBCI), which reframes causal effect estimation as a dataset-level prediction task. Instead of relying on manually designed estimators tailored to specific causal settings—such as regression-based, propensity score, or instrumental variable methods—BBCI learns to predict causal effects by training on sampled dataset-effect pairs. The authors demonstrate that BBCI can accurately estimate both average treatment effects and conditional average treatment effects across a range of identified causal inference scenarios, including settings where traditional estimators are less developed. This work offers a unified, learning-based alternative to the design of bespoke causal estimators.

**Audience:**

Yes

**Claims And Evidence:**

Yes

**Requested Changes:**

**1.** The notion of a *causal query* $q$ is referenced multiple times but not formally defined. The authors should provide a precise and complete explanation of what constitutes a causal query in their framework.

**2.** It is unclear whether the covariates $x$ used in the model form a valid adjustment set. Are these covariates assumed to be sufficient for adjustment, or does the method involve selecting an adjustment set during training? The authors should clarify this point.

**3.** In scenarios with unobserved confounding—such as in instrumental variable (IV) models or proxy variable models—it is unclear how a "true" structural causal model (SCM) can be constructed or specified. The authors should clarify their assumptions and discuss the feasibility of SCM specification in such settings.

**4.** The problem formulation seems to assume a uniquely identifiable model when constructing the predictive mapping from datasets to causal effects. The authors should discuss how their method performs in cases where the causal query is not uniquely identifiable, and whether the learned model accounts for such ambiguity.

**5.** Given a sample, it is not clear how the distribution and associated parameters (denoted $S_v$) are determined. The authors should elaborate on how these are inferred from data or otherwise specified.

**6.** In Algorithm 1, the function class $f_\theta$ is not formally defined. The authors are encouraged to specify the architecture or hypothesis space used for this predictive model.

**7.** In Step 5 of the procedure, the meaning of $f_\theta(\tilde{D})$ remains unclear. Does this quantity correspond to a causal effect estimated using an external method or ground-truth simulator? Its precise definition and the implementation approach should be made explicit.

**8.** The experimental settings described in Sections 3.1 and 3.2 differ from those in Section 3.4, but the nature of these differences is not clearly explained. The authors should clarify how the experimental setups vary, including any differences in identification assumptions, data generation, or evaluation criteria.

**Strengths And Weaknesses:**

Strengths:

**1.** The paper tackles the important and widely relevant problem of causal effect estimation, which has critical applications in domains such as medicine and economics. The practical significance of the problem lends weight to the study.

**2.** The authors conduct comprehensive experiments across a variety of settings, including both scenarios with established estimators and those lacking them, comparing the proposed BBCI method against a broad spectrum of existing approaches.

Weaknesses:


**1.** The setting adopted in the paper may lack practical applicability. For instance, in the presence of unobserved confounders, it is unclear how a true structural causal model (SCM) can be reliably constructed or specified.

**2.** Several parts of the paper suffer from unclear or ambiguous descriptions, which hinder the reader’s understanding of the proposed methodology.

**3.** Some notations are introduced without sufficient explanation or formal definitions, making it difficult to follow the algorithm and its theoretical underpinnings.

---

> ### Author Response · Authors · 2025-05-26
>
> Thank you for the review, comments, and suggestions for improvement. We discuss each of the requested changes/clarifications below.
> 1.  “The notion of a causal query $q$ is referenced multiple times but not formally defined. The authors should provide a precise and complete explanation of what constitutes a causal query in their framework.”
>     - A causal query is any functional of the SCM with a certain variable intervened on with different values. For example, the ATE is the difference between the average outcome with the treatment intervened on with two different values.
> 2. “It is unclear whether the covariates $x$ used in the model form a valid adjustment set. Are these covariates assumed to be sufficient for adjustment, or does the method involve selecting an adjustment set during training? The authors should clarify this point.”
>     - BBCI takes as input all variables that are available. Given identification, BBCI learns to use whatever features are sufficient for effect estimation. Specifying the adjustment set is unnecessary.
> In order to test how well BBCI performs in different settings against a known ground truth, we create scenarios where some variables are hidden from BBCI (e.g., a confounder is hidden), but this is not specifying an adjustment set for the model; rather, it is just creating the known ground truth.
> 3. “In scenarios with unobserved confounding—such as in instrumental variable (IV) models or proxy variable models—it is unclear how a "true" structural causal model (SCM) can be constructed or specified. The authors should clarify their assumptions and discuss the feasibility of SCM specification in such settings.”
>     - In settings with a real observed dataset, where the true SCM is unknown, we don’t necessarily need to construct feature relationships. We can condition on the observed dataset. For example, in Section 5, we use the features as-is and simulate only the response surface. The same can be done in the IV and proxy settings, rather than trying to reconstruct unobserved confounders for the given real dataset. As mentioned above, we simulate data in other examples in order to show efficient estimator development in known settings and to test how well BBCI performs in different settings of interest with a known ground truth.
> 4. “The problem formulation seems to assume a uniquely identifiable model when constructing the predictive mapping from datasets to causal effects. The authors should discuss how their method performs in cases where the causal query is not uniquely identifiable, and whether the learned model accounts for such ambiguity.”
>     - We agree with this point, and the current approach indeed produces point estimates and would not automatically account for ambiguity in the effect. Adding uncertainty so that we get interval and/or distributional rather than point estimates of the effect is an area of immediate interest for future work.
> 5. Given a sample, it is not clear how the distribution and associated parameters (denoted $S_v$) are determined. The authors should elaborate on how these are inferred from data or otherwise specified.
>     - We believe this point is related to the same confusion in (3) — we do not need to infer parameters from data in order to use BBCI.
> 6. In Algorithm 1, the function class $f_\theta$ is not formally defined. The authors are encouraged to specify the architecture or hypothesis space used for this predictive model.
>     - In the current work, we define $f_\theta$ as any function class that takes variable-length datasets as input and outputs a scalar. We use permutation-invariant models as a way to further reduce the hypothesis space, because we know the causal estimands we consider are all permutation-invariant. We thus use SetTransformer++ [Zhang et al. 2022] as an example of a permutation-invariant model throughout our experiments.
> 7. In Step 5 of the procedure, the meaning of $f_\theta(\tilde{D})$ remains unclear. Does this quantity correspond to a causal effect estimated using an external method or ground-truth simulator? Its precise definition and the implementation approach should be made explicit.
>     - $f_\theta(\tilde{D})$ refers to an $f_\theta$, that is being learned, applied to the dataset $D$. The other term, $\phi(K; S_\nu)$, is defined in Step 4.
> 8. The experimental settings described in Sections 3.1 and 3.2 differ from those in Section 3.4, but the nature of these differences is not clearly explained. The authors should clarify how the experimental setups vary, including any differences in identification assumptions, data generation, or evaluation criteria.
>     - The data generation settings for 3.4 are the same as the ones used in 3.1 and 3.2 in terms of identification settings and evaluation criteria. The main differences are the training hyperparameters such as model size and epochs (128 vs. 192 embedding dimension, 256 vs. 32 inducing points, 50 vs. 200 training steps), which we will add to the text description.

---

### Decision · Action_Editor_ehn5 · 2025-06-22

**Recommendation:** Reject

**Additional Comments:**

The paper is not at the adequate level.

**Audience:**

Yes

**Audience Explanation:**

The topic is clearly of wide interest, and the claims made by the authors have attracted the attention of reviewers. This is why there is potential to improve the paper.

**Claims And Evidence:**

No

**Claims Explanation:**

While there is merit in this work, the reviewers agree that it is not at an adequate level for TMLR. Causal claims require strict consideration of identifiability conditions, which are not adequately addressed in the manuscript.
The experimental settings used by the authors are too simple to allow for meaningful conclusions to be drawn. The claims made in the paper about the generality of the proposed approach are also problematic.
This means that more thorough writing and experiments would make the paper worthy of publication.

More in detail, the authors haven't done a good job at addressing the concerns of Reviewer Mebz's, which raised doubts for the other reviewers. Specifically, despite the authors' claims, this paper does not succeed in its stated goal of providing a general-purpose method for estimating causal effects in settings where identification holds, especially in the absence of well-established estimators.
* In particular, the papers still suffers from a lack of attention to identifiability: The proposed method assumes that identification holds but does not incorporate or test the necessary assumptions. In fact, it treats identifiable settings as if they were unidentifiable, e.g., failing to leverage essential non-structural knowledge.
* Trivial experimental settings: The empirical evaluations are conducted in trivial settings (e.g., no meaningful finite-sample challenges) and rely on simulators with known SCMs, undermining the claim that the method works in settings without established estimators.
* The papers suffers from overstated claims about generality and estimator development : The method requires tailoring to specific graphical structures, which contradicts its promise to simplify estimator development or generalize across settings.
* The manuscript has not been properly revised.
Overall, while there is merit to this work, but it needs more work and thinking on the raised concerns. The core issues relate to a fundamental understanding of what makes causal inference hard: 1) identification and the assumptions that support it, and 2) specific estimator designs for finite sample efficiency. The authors have not addressed these challenges in either the method or the response.

**Resubmission Of Major Revision:**

The authors may consider submitting a major revision at a later time.